# Phosphatidylinositol-4-phosphate controls autophagosome formation in Arabidopsis thaliana

Rodrigo Enrique Gomez[1], Clément Chambaud ®[1], Josselin Lupette ®[1], Julie Castets[1], Stéphanie Pascal ®[1], Lysiane Brocard[2], Lise Noack[3], Yvon Jaillais ®[3], Jérôme Joubès ®[1] & Amélie Bernard ®[1] ✉

Autophagy is an intracellular degradation mechanism critical for plant acclimation to environmental stresses. Central to autophagy is the formation of specialized vesicles, the autophagosomes, which target and deliver cargo to the lytic vacuole. How autophagosomes form in plant cells remains poorly understood. Here, we uncover the importance of the lipid phosphatidylinositol-4-phosphate in autophagy using pharmacological and genetical approaches. Combining biochemical and live-microscopy analyses, we show that PI4K activity is required for early stages of autophagosome formation. Further, our results show that the plasma membrane-localized PI4Kα1 is involved in autophagy and that a substantial portion of autophagy structures are found in proximity to the PI4P-enriched plasma membrane. Together, our study unravels critical insights into the molecular determinants of autophagy, proposing a model whereby the plasma membrane provides PI4P to support the proper assembly and expansion of the phagophore thus governing autophagosome formation in Arabidopsis.

Macroautophagy (hereafter referred to as autophagy) is an evolutionary conserved recycling mechanism during which intracellular material including proteins, proteins aggregates, organelles, and/or invading pathogens are delivered to the lytic vacuole for degradation[1]. Autophagy exists at a basal level in optimal environmental conditions and is strongly induced by a wide range of stresses during which it plays essential roles in plant physiology by (1) clearing the cells from unwanted or toxic components thus ensuring quality control and (2) recycling organelles/molecules that can be re-used by the cell to support or re-shape metabolism in periods of stress[2]. As such autophagy is a critical component of plant acclimation to environmental changes and plants lacking autophagy are unable to survive to various conditions including nitrogen starvation, drought, or pathogen attacks[3].

Upon induction, autophagy morphologically starts with the de novo nucleation of an initial membrane structure called the phagophore which then expands around autophagy cargo by the addition of lipids/membranes and ultimately seals to form a double membrane vesicle called autophagosome (AP)[4]. Upon completion, the AP traffics to the vacuole where the outer membrane of the AP fuses with the tonoplast releasing the inner membrane and the cargo into the vacuolar lumen where they are rapidly degraded by hydrolases. The molecular machinery of autophagy is governed by a conserved core of ATG (AuTophaGy) proteins[5,6]. At steady state most ATG proteins reside in the cytosol; upon induction of autophagy, they are rapidly and sequentially recruited to a lipido-proteic core named the pre-autophagosomal structure (PAS) which serves as the nucleation site for the phagophore. At the PAS, the ATG machinery regulates the assembly of the phagophore in a spatio-temporally coordinated manner. Upon induction, the ATG1/ATG13 complex allows the recruitment of an autophagy-specific

[1]Laboratoire de Biogenèse Membranaire, UMR 5200, CNRS, Univ. Bordeaux, F-33140 Villenave d'Ornon, France. [2]Plant Imaging Platform, Bordeaux Imaging Center, UMS 3420, CNRS, Univ. Bordeaux, F-33140 Villenave d'Ornon, France. [3]Laboratoire Reproduction et Développement des Plantes, Univ. Lyon, ENS de Lyon, CNRS, INRAE, F-69342 Lyon, France. ✉e-mail: amelie.bernard@u-bordeaux.fr

phosphatidylinositol 3-kinase (PI3K) complex which promote the phagophore nucleation[7]. The local enrichment of PI3P at the early autophagy structure is required for the recruitment of downstream ATG proteins including ATG18, ATG2 as well as the ATG12-ATG5-ATG16 complex. Notably, the ATG12-5-16 complex participates in the ubiquitin-like conjugation machinery which mediates the incorporation of ATG8 into the phagophore membrane by covalently lipidating the protein to a phosphatidylethanolamine[8] (PE). The lipidation of ATG8 is a central and finely regulated step in autophagy which (1) supports the expansion of the phagophore membrane and (2) controls the recognition of cargo during selective autophagy[9]. Mutants defective for ATG8 conjugation are unable to form mature APs[10,11].

APs are central structures in autophagy, targeting and delivering cargo, thus modulating the rate of autophagy degradation and plant responses to stress; nevertheless, the molecular mechanisms underlying their formation remain largely unknown in plants. AP biogenesis relies on extensive membrane remodeling events governing the nucleation, assembly, and structuration of the cup-shaped phagophore membrane, its expansion and ultimate fusion to form a complete double membrane vesicle[12]. Strikingly, while AP formation is a membrane-based process, very little is known about the nature and functions of lipids in plant autophagy besides the importance of PI3P and ATG8 conjugation[13,14]. Yet, lipids are critical components of biological membrane, both by providing building blocks for its formation and by defining membrane properties. Physico-chemical properties of lipids, their distribution, and their interconversion in membranes are critical determinants of the identity, the structure, the functions and the remodeling of membranes, promoting curvature, fusion, or fission events as well as the establishment of contacts between membranes[15,16]. In that context, we postulated that lipids have the potential to instruct fundamental aspects of AP formation.

Upstream of the present study, we performed a lipid inhibitor screen to identify and functionally characterized lipids that are critically engaged in autophagy in Arabidopsis. As presented here, our results showed that pharmacologically or genetically inhibiting PI4P synthesis blocks autophagy in plant cells. By combining biochemical and live-microscopy analyses, we showed that PI4K activity is required for AP formation. The absence of PI4P hampered the lipidation of ATG8 but did not prevent the membrane-association of earlier components of the autophagy machinery suggesting that PI4K controls the proper assembly and elongation of the phagophore. Our results further show that a substantial portions of autophagy structures are found in proximity to PI4P-enriched compartments and that the plasma membrane (PM)-localized PI4Kα1 is involved in autophagy thus proposing a model in which the PM may provide PI4P to support AP formation in Arabidopsis.

## PAO, an inhibitor of PI4K activity, prevents autophagic flux

To identify which and how lipids affect the autophagy process in plants, we performed a lipid inhibitor screen. We treated Arabidopsis seedlings with a range of inhibitors known to affect the synthesis of specific lipids and measured the impact of these inhibitors on autophagy activity upon nutrient starvation conditions using the GFP-ATG8 assay. During autophagy, ATG8 is recruited to the membrane of the phagophore, the precursor to the AP, and a portion of the protein is delivered in the vacuole after fusion of the AP with the tonoplast. In the vacuolar lumen, ATG8 proteins fused to the GFP (GFP-ATG8) are rapidly degraded while the more stable GFP moiety accumulates. Therefore, the ratio between GFP-ATG8 and free-GFP reflects the rates of autophagy cargo delivery[17]. Seedlings were treated with the PI4K inhibitor PAO (phenylarsine oxide) at a range of concentrations previously described[18–21]. Upon treatments, roots were dissected and their proteins were analyzed by western blot. We observed a large decrease in the accumulation of free GFP using PAO compared to control

conditions after 1.5 or 3 h of nutrient starvation suggesting a block in autophagy activity in these conditions (Fig. 1a–d, compare PAO to DMSO). To validate that PAO treatment was effectively inhibiting the synthesis of PI4P in our conditions, we performed confocal microscopy analysis on plants expressing the fluorescent probe mCitrine-PH^OSBP, a PI4P reporter[18,22]. Under nutrient starvation conditions the fluorescence of the PI4P reporter mostly localizes at the PM (Supplementary Fig. 1a, c, DMSO); in contrast, PAO treatment resulted in diffuse fluorescence of the probe throughout the cytosol (Supplementary Fig. 1a, c, PAO) indicating that PI4P synthesis was prevented (as reported in ref. 18).

Target of Rapamycin (TOR) is a conserved master regulator of autophagy[23–25]. To investigate if the synthesis of PI4P was required for TOR-dependent autophagy induction, we used the TOR inhibitor AZD8055[26,27] (AZD) in combination with PAO. Upon treatment with 1 μM AZD, we observed a massive accumulation of free GFP and an almost total disappearance of GFP-ATG8, indicative of a large increase in autophagy flux when TOR activity is inhibited (Fig. 1e, f). In contrast, when plants were treated with both AZD and PAO, the degradation of GFP-ATG8 to free GFP was largely stalled in a dose-dependent manner (Fig. 1e, f) indicating that PAO blocks autophagy flux in these conditions and suggesting that PI4K activity is required for TOR-induced autophagy activation. Altogether, our results show that PAO is a potent autophagy inhibitor and suggest that PI4P synthesis is critical for autophagy in root cells.

## Downregulating *PI4Kα*, rather than *PI4Kβ*, decreases autophagic degradation

Our results suggest that PI4P synthesis is required for autophagy. In Arabidopsis, three phosphatidylinositol-4-kinases (PI4-Kinases), PI4Kβ1, PI4Kβ2, and PI4Kα1 have been characterized as bona fide PI4P producing enzymes[28–30]. PI4Kβ1 and PI4Kβ2 both localize to the trans-Golgi network/early endosomes (TGN/EE)[31–34]; while PI4Kα1 mostly localizes at the PM[30]. To genetically confirm the result obtained using PAO as well as exploring the subcellular origin of PI4P for autophagy, we compared the relative contribution of PIKα1 and PI4Kβ in autophagy activity using reverse genetics. We used the previously described *pi4kβ1 pi4kβ2* double knock-out Arabidopsis mutant[31,32]; however, no homozygous tDNA-insertion mutant line for *PI4Kα1* was available as PI4Kα1 is required for pollen development[30,35]. We thus generated an artificial microRNA (amiRNA) line designed against PI4Kα and placed under the control of a β-estradiol inducible promoter. After 24 h of β-estradiol treatment, the *amiRNA:PI4Kα1* Arabidopsis line showed a decrease of up to 60–80% of *PI4Kα1* mRNA (Supplementary Fig. 2a, b).

Autophagy has been consistently shown to promote plant survival upon nutrient limitation[36]. To assess the function of PI4 kinases in plant physiology, we thus compared the recovery of the *amiRNA:PI4Kα1* and the *pi4kβ1 pi4kβ2* lines to that of WT plants as well as of the *atg5-1* mutant after nutrient starvation. Briefly, 7-day-old seedlings were transferred from rich MS plates (+N) to MS plates lacking nitrogen and carbon (−NC) and placed in darkness for 7 days to induce autophagy. Plants were then transferred back to +N MS plates for 10 days to recover (see Supplementary Fig. 2c for additional explanation of the experimental procedure). As seen in Fig. 2a, after 10 days of recovery post starvation in β-estradiol conditions, most *amiRNA:PI4Kα1* seedlings displayed a senescence-like phenotype similar to the one observed in *atg5-1* lines with a median survival rate of 7% largely inferior to that of WT plants of around 70% (Fig. 2a). As previously described, the *pi4kβ1 pi4kβ2* double mutant showed delayed development with shorter root than WT (due to defects in cell expansion and cytokinesis) even in +N conditions[31]. Nevertheless, our experiments showed no significant difference in the survival rates of *pi4kβ1 pi4kβ2* compared to WT (Fig. 2a). Together, these results indicate that PI4Kα1, but not PI4Kβs, are required for plant acclimation to nutrient starvation, suggesting that PI4Kα1 is involved in autophagy.

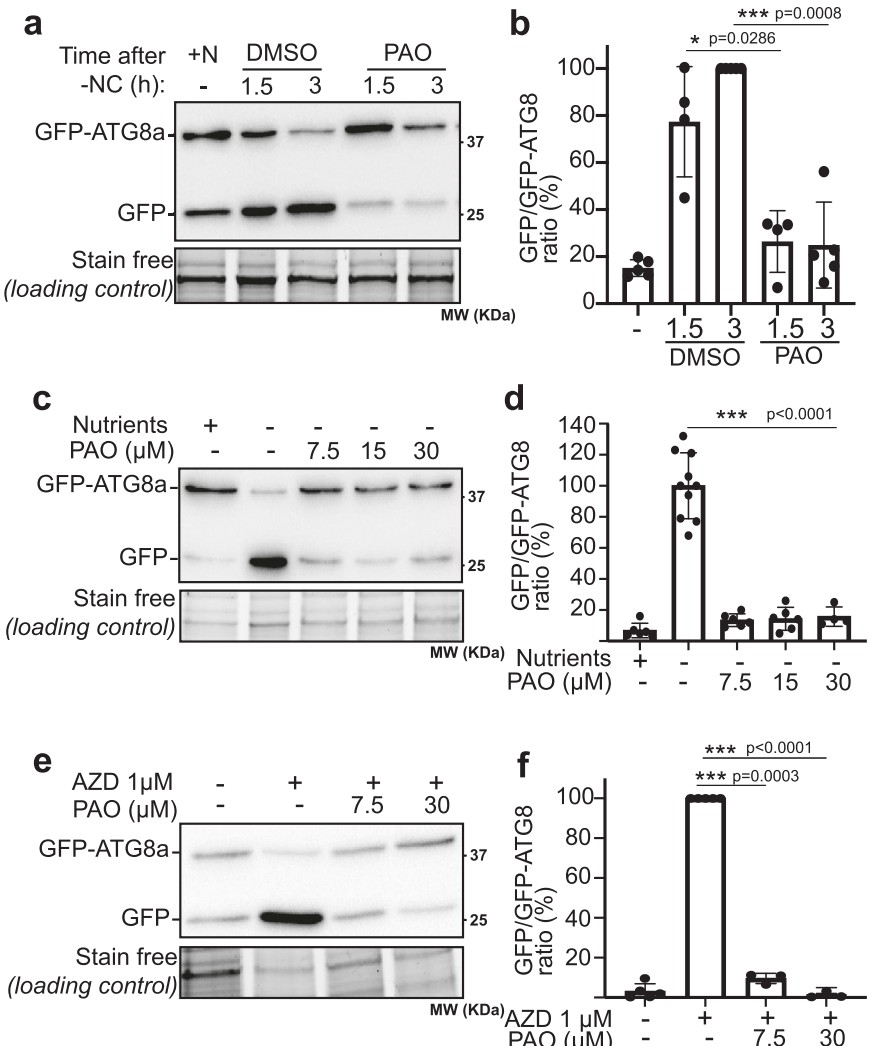

**Fig. 1 | The PI4-kinase inhibitor PAO blocks autophagic flux in root cells.**
**a**, **c**, **e** Detection of GFP-ATG8a degradation and release of free GFP in the vacuole. 7-day-old GFP-ATG8a seedlings were transferred from nutrient rich MS plates to either nutrient rich liquid medium (+N), nutrient starved liquid medium (-nitrogen and carbon; −NC) or nutrient rich liquid medium supplemented with 1 μM AZD in either PI4K inhibiting conditions (+PAO) or control conditions (DMSO). Roots were dissected from seedlings after treatments and total root proteins were extracted and subjected to immunoblot analysis with anti-GFP antibodies. Stain free images were used as loading control. Uncropped blots in Source Data. In **a**, seedlings were transferred to nutrient rich liquid medium for 3 h (+N) or in nutrient deprived conditions and treated with 30 μM PAO or DMSO as control for 1.5 h or 3 h. In **c**, seedlings were treated for 3 h with various concentrations of PAO as indicated on the figure. In **e**, seedlings were transferred to liquid medium containing DMSO or 1 μM AZD for 3 h supplemented with PAO at either 7.5 μM or 30 μM, or DMSO as control. **b**, **d**, **f** Quantification of the ratio of GFP/GFP-ATG8a reflective of the rate of autophagy flux in each experimental conditions presented in **a**, **c**, and **e** respectively, relative to the ratio of control plants (−N+DMSO 3 h or + AZD+DMSO) which was set to 100% in each independent experiments. Results are presented as the mean ± SD with values of distinct replicates (**b**, number of independent experiments: $n = 4$ for 1.5 h conditions, $n = 5$ for other conditions; **d**, $n = 6$ distinct replicates examined over four independent biological experiments; **f** number of independent biological experiments: $n = 3$ for PAO conditions, $n = 5$ for other conditions) with two-tailed one sample t-test (for comparison to control at 100% in **b**, **d**, **f**) or two-tailed Mann-Whitney test (compare DMSO 1.5 h to PAO 1.5 h in **b**).

To further test this idea, we assessed the implication of PI4Kβ and PI4Kα1 in autophagy activity, by comparing the rate of degradation of the autophagy adaptor protein NBR1 in these plants, a well-described marker of autophagy flux[37]. As seen in Fig. 2b, c, in WT plants NBR1 is massively degraded upon autophagy induction by nutrient starvation. The addition of concanamycin A, which de-acidifies the vacuole resulting in a reduction of vacuolar degradation[17], partially inhibits NBR1 degradation. In contrast, the protein highly accumulates in the autophagy deficient *atg5-1* mutant in all conditions, with no significant differences in protein level upon autophagy induction. In the same conditions, the *pi4kβ1 pi4kβ2* double mutant showed no statistical difference in regard to the level of NBR1 compared with the WT (Fig. 2b, c, left panel). In contrast, the *amiRNA:PI4Kα1* line showed a slight, albeit significant, over-accumulation of the protein compared to the WT upon autophagy induction by nutrient starvation (Fig. 2b, c,

right panel) suggesting that NBR1 degradation is altered upon PI4Kα1 inhibition. Note that mRNA analyses showed no impact of knocking out *PI4Kβ* or knocking down *PI4Kα1* in *NBR1* expression across conditions (Fig. 2d), supporting that the differences in NBR1 protein levels are due to modified rates of protein degradation rather than differences in gene expression. Together these results suggest that PI4Kα1, rather than PI4Kβ, is preferentially required to support proper autophagy degradation.

## PI4K activity is essential for AP formation
Our results show that autophagic flux is blocked following the inhibition of PI4K; we thus decided to explore the role of PI4P in autophagy, i.e., the step(s) of the pathway at which this lipid may function. To this end, we used confocal microscopy to visualize autophagy structures marked by GFP-ATG8 in vivo in the roots of Arabidopsis seedlings.

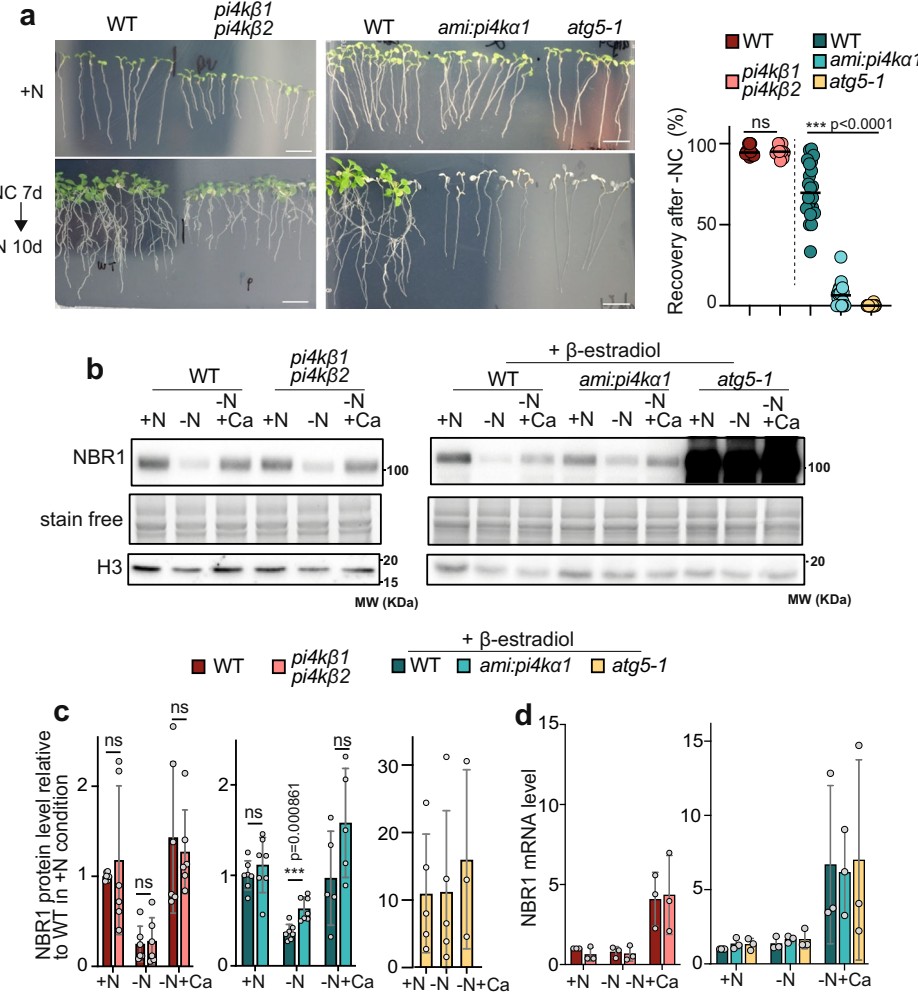

**Fig. 2 | Downregulating *PI4Kα1*, but not *PI4Kβ*, compromises autophagy activity. a** *amiRNA:PI4Kα1* seedlings display phenotypes reminiscent of an autophagy deficiency after recovery from nutrient starvation conditions. An overview of the experiment is presented in Supplementary Fig. 2c. Briefly, 7-day-old seedlings grown in rich conditions (+N) were transferred to nutrient deprived (−NC) solid medium for 7 days in darkness and then transferred back to +N solid medium for 10 days for recovery. Media were supplemented with 10 μM β-estradiol to compare the survival of *amiRNA::PI4Kα1* to that of WT and *atg5-1*. Scale Bar: 2 cm. To assess the percentage of recovery presented in the right panel, three independent experiments were carried out for PI4Kβ analyses with a total of *n* = 12 distinct replicates; four independent experiments were carried out for PI4Kα1 analyses with a total of *n* = 22, 22, or 24 distinct replicates for WT, *amiRNA:PI4Kα1* or *atg5-1*, respectively. Statistical differences were assessed using two-tailed unpaired t-test, exact *p* values are provided in the source data file. **b** Immunoblot analyses of NBR1 protein levels from 7-day-old seedlings. An overview of the experimental set up is presented in Supplementary Fig. 2b. Uncropped blots in Source Data. Col-0 (WT), *amiRNA:PI4Kα1*, *atg5-1*, were grown on full MS agar plates, then transferred to β-estradiol containing MS plates for 24 h before being transferred in liquid nutrient rich (+N), liquid nutrient starvation medium (−N), or liquid nutrient starvation medium supplemented with 1 μM concanamycin A (+Ca), for 16 h. In the other set of

samples, Col-0 (WT) and *pi4kβ1 pi4kβ2* double mutant plants were directly transferred from full MS plates to liquid rich, liquid nutrient starvation or liquid nutrient starvation media supplemented with 1 μM concanamycin A (+Ca) for 16 h. **c** Quantification of the NBR1 protein levels relative to that of WT plants in +N conditions (set to 1) in the lines and conditions presented in (**b**). Signal from stain free gels and Histone H3 bands were used as loading control and for normalization. Results present the average ± SD and individual values. For PI4Kα1 experiments, in +N and −N conditions, *n* = 7 biological replicates were examined over four independent experiments for WT and *amiRNA:PI4Kα1*, *n* = 5 biological replicates in 3 independent experiments for *atg5-1*; in −N + Ca conditions, *n* = 5 biological replicates in three independent experiments for WT and *amiRNA:PI4Kα1*, *n* = 3 independent experiments for *atg5-1*. For PI4Kβ experiments, *n* = 6 biological replicates were examined over three independent experiments. Statistical differences were assessed using two-tailed unpaired t-test. ns, non significant. **d** *NBR1* mRNA levels in the line and conditions presented in (**b**). Results represent the average ± SD of *n* = 3 independent experiments. Statistical differences were assessed using two-tailed one sample t-test (for comparison to WT in +N condition which was set to 1 in each experiment) or two-tailed Mann–Whitney test (for comparison to WT in −NC condition or WT in −NC+Ca conditions). Statistical analyses showed no significant differences across lines and conditions.

Plants were analyzed in rich conditions as well as after 30 min of nutrient starvation +/− PAO, conditions previously shown to sharply disrupt PI4P synthesis without affecting either the pool of $PI(4,5)P_2$ (ref. 18 and Supplementary Fig. 1a, b, e, f) or cells/plant viability (Supplementary Fig. 3; Supplementary Movies 1−3,). Upon 30 min of transfer in liquid medium lacking nutrients in darkness, the number of GFP-ATG8a puncta significantly increased compared to plants in +N conditions which reflects an increase in AP formation (Fig. 3,

compare +NC and −NC, DMSO conditions; Supplementary Fig. 4a, b). In comparison, when plants were treated with PAO in the same conditions, very little GFP-ATG8 puncta were detected and GFP-ATG8a was mostly found in the cytosol suggesting that AP formation was prevented (Fig. 3). Similarly, treatment with PAO greatly affected the number of GFP-ATG8a puncta upon autophagy induction by AZD compared to control conditions (Supplementary Fig. 4c, d), suggesting that PI4K activity is critical for TOR-

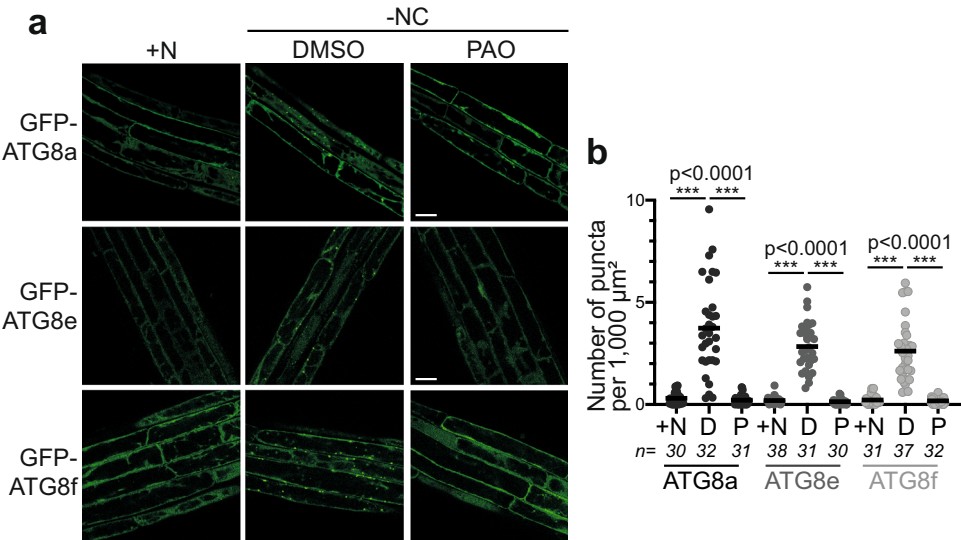

**Fig. 3 | Inhibition of PI4K activity blocks autophagosome formation. a** Confocal images of roots from Arabidopsis plants expressing different isoforms of ATG8: GFP-ATG8a, GFP-ATG8e or GFP-ATG8f. 7-day-old plants were imaged in either nutrient rich conditions (directly from MS plates; +N) or after 30 min in nutrient deprived liquid medium (−NC) with the addition of PAO (P, 60 μM) or DMSO as control (D). Scale Bar: 20 μm. **b** Quantification of autophagic structures in conditions presented in (**a**). Results are presented as number of puncta per 1000 μm² of root area, three independent experiments were performed for each condition, number of replicates (*n*, images) are indicated in the figure. For each ATG8 isoform, one-way ANOVA analyses showed statistical differences among the three conditions (+N, D, P, *p* < 0.0001 for each ATG8, see source data file for exact *p* values). Two-tailed t-test with a Bonferroni correction was used as a post hoc test showing statistical differences when +N or P conditions were compared to D; exact *p* values are provided in the source data file.

dependent AP formation. The Arabidopsis genome contains nine genes encoding ATG8 isoforms[38], and specific ATG8 isoforms decorate specific population of APs depending on the stress responsible for autophagy induction[9,39]. It has been hypothesized that ATG8 specialization contributes to the differentiation of selective autophagy pathways (i.e., different ATG8s isoforms interact with specific cargo receptors[39]). To further explore the importance of PI4P synthesis for AP formation, we analyzed the impact of PAO on the subcellular localization of additional ATG8 isoforms, namely ATG8e and ATG8f. Similar to what we observed for ATG8a, our results showed that PAO treatment largely blocks the formation of GFP-ATG8e or GFP-ATG8f punctae upon nutrient starvation conditions (Fig. 3). Together, these analyses show that PAO inhibits the formation of ATG8-labeled structures in multiple autophagy inducing conditions suggesting that PI4K activity is essential for AP formation.

## PI4K activity controls ATG8 lipidation

A prerequisite for the formation of APs, is the recruitment of the central protein ATG8 to the phagophore membrane[40] which requires the covalent conjugation of ATG8 to the lipid PE. Our results showed that PI4K activity is required for the formation of ATG8-labeled structures (Fig. 3) implying that AP formation is prevented and raising the question of ATG8 recruitment to the phagophore in these conditions. To address this question, we tested the effect of blocking PI4K on ATG8 lipidation. Roots were dissected from WT plants in either rich conditions (R, from seedlings grown on +N MS plates), or after 30 min in nutrient-starved liquid medium in darkness with either PAO (P) or DMSO (D) as control (Fig. 4a, b). We performed cell fractionation on dissected roots by using high-speed ultracentrifugation to separate soluble proteins (S100) and membrane-bound proteins (P100). Immunoblot analyses using an endogenous anti-ATG8 antibody showed that the quantity of ATG8 was similar in total lysates or in the soluble fractions across treatments (Fig. 4a, b) suggesting that PAO has no effect on the production of the ATG8 protein. In the membrane fraction (P100), we observed a moderate increase in ATG8-PE in

nutrient-starved samples in control conditions (DMSO) compared to rich conditions (Fig. 4a, b). Additional experiments showed that this increase likely results from nutrient starvation rather than from the stress provoked by transferring seedlings from solid to liquid medium (Supplementary Fig. 5a, b). When compared to the DMSO control conditions, the level of the ATG8-PE band was significantly reduced in the membrane fraction upon PAO treatment (Fig. 4a, b, ATG8-PE) suggesting that the lipidation of ATG8 is prevented in the absence of PI4K activity. We further assessed ATG8 lipidation in the *PI4Kα1* or *PI4Kβ* deficient lines by performing similar cell fractionation analyses. To assess the impact of *PI4Kα1*, WT and *amiRNA:PI4Kα1* seedlings were first transferred from +N MS plates to +N liquid medium supplemented with β-estradiol for 24 h. Seedlings were then transferred to nutrient-deprived liquid medium for various times as indicated in Supplementary Fig.5c. In WT plants, we observed a significant increase in ATG8-PE shortly after autophagy induction (Supplementary Fig. 5c, d, 0.5 h), indicating a boost in ATG8 lipidation. This is followed by a reduction in the level of the protein after 3 h that is prevented by the addition of concanamycin A, and thus reflects the rate of ATG8-PE degradation by autophagy (Supplementary Fig. 5d, e; see ATG8-PE index calculated as the level of ATG8-PE in conditions 3+Ca/level of PE at 3 h of −NC). The levels and variations of ATG8-PE were similar to that of WT in the *pi4kβ1 pi4kβ2* mutant with no significant difference in any time points or conditions indicating that ATG8 lipidation and degradation were unchanged in the mutant. In contrast, the level of ATG8-PE did not significantly increase in the *amiRNA:PI4Kα1* line after 0.5 h of nutrient starvation compared to +N or compared to WT in the same conditions, suggesting a reduction in the rate of ATG8 lipidation. In addition, we found that after 3 h of starvation, the level of ATG8-PE was mostly unchanged compared to +N or − 0.5 h conditions, which translated in a mild but significant reduction in the average of the ATG8-PE index compared to that WT and thus indicates a defect in ATG8 degradation (Supplementary Fig. 5d, e). These results suggest that both the conjugation and degradation of ATG8-PE are hampered in the *amiRNA:PI4Kα1* line which further support a role for this enzyme in the autophagy pathway.

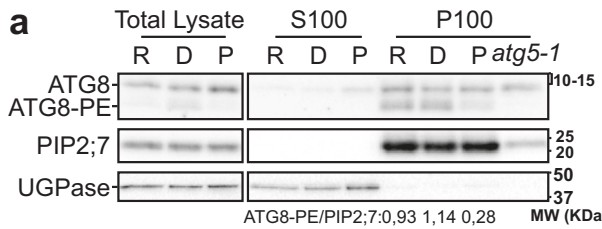

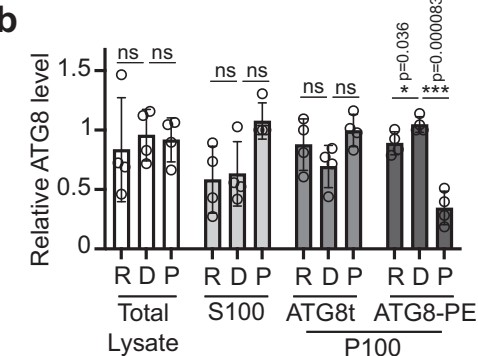

Fig. 4 | PI4K inhibition blocks ATG8 lipidation. a Immunoblot analysis using anti-ATG8 antibody of WT or *atg5-1* roots in rich conditions (R) or after 30 min in nutrient depleted conditions (−NC) with PAO 60 µM (P) compared to control conditions (D, DMSO). Uncropped blots in Source Data. Lipidated ATG8 is present in the pellet membrane containing fraction (P100), while being excluded from the supernatant (S100) fraction as shown by the comparison of the protein profiles between WT and *atg5-1* mutant in which ATG8 conjugation is prevented. PIP2;7 was used as loading control and normalization factor for the membrane fraction; UGPase was used as loading control and normalization factor for the soluble fraction. Levels of ATG8-PE normalized to PIP2;7 in P100 of the selected blot is reported. b Quantification of ATG8 levels in conditions presented in (a). Results present the average of the level of ATG8 and ATG8-PE band intensities relative to each type of sample and normalized to either PIP2;7 or UGPase ± SD as well as individual values (n = 4 independent experiments) with two-tailed paired t-test to compare between Rich and DMSO or DMSO and PAO conditions. ns, non-significant.

## PI4P synthesis is critical for the proper assembly/elongation of the phagophore

Our results indicate that the formation of the mature autophagy structure, the AP, is prevented in the absence of PI4P; we next wondered if an earlier autophagic structure could still be formed in these conditions. To address the function of PI4P in the formation of the phagophore, we focused on proteins that are only present on phagophores but dissociate from completed APs. Further, because our results show that ATG8 lipidation is prevented in the absence of PI4P, we chose to focus on proteins which recruitment to the phagophore precedes ATG8 lipidation at this membrane. In other organisms, both the initiation ATG1 complex as well as ATG9 proteins act upstream and are required for the lipidation of ATG8[41]. In Arabidopsis, however, the *atg13* (a member of the initiation complex) or *atg9* mutants do not display a default in the lipidation of ATG8, indicating that these proteins are dispensable for this step[42,43]. Thus, we focused on the ATG5 and the ATG12-ATG5 conjugate, two essential components of the ATG8 lipidation complex[11] (Fig. 4a, *atg5-1*). We reasoned that if the absence of PI4P synthesis blocks the overall autophagy pathway -with no nucleation of the PAS/early phagophore- ATG12-ATG5 would remain in the cytosol and thus be detected in the soluble fraction and depleted in the membrane fraction upon cell fractionation, similar to what we found for ATG8 (Fig. 4). To test this hypothesis, we extracted proteins from roots of ATG5-mCherry expressing plants and performed cell fractionation as described above. Both ATG5-mCherry and ATG12-ATG5-mCherry showed the same levels in all conditions tested, indicating

that PAO treatment affected neither ATG5-mCherry protein production nor its conjugation with ATG12 (Fig. 5a, b). However, we observed that both ATG5-mCherry and ATG12-ATG5-mCherry accumulated at the membrane fraction from +NC to -NC treatment in PAO conditions compared to control DMSO-treated plants (Fig. 5a, b). ATG5 and ATG12-ATG5 are soluble and reside in the cytosol at steady state. Upon autophagy induction, they are rapidly recruited to the membrane of the phagophore/PAS prior to the recruitment and lipidation of ATG8 at this membrane. In that context, our results showing an accumulation of ATG12-ATG5 in membranes upon PAO indicate that PI4P is not required for the recruitment of ATG5 or ATG12-ATG5 suggesting that early events of autophagy are still occurring in the absence of PI4K activity with notably the nucleation of an early phagophore/PAS structure. This hypothesis was further supported by our analyses of two additional ATG proteins: ATG4, a cysteine protease that cleaves the N-terminal Glycine residue of ATG8, a prerequisite for ATG8 lipidation[10,44] and ATG18, a PI3P-binding protein critical for autophagy in Arabidopsis[43,45] and which was recently suggested to tether the PAS to the ER, in complex with ATG2, to initiate membrane expansion during AP formation in yeast[46]. Similar to what we found for ATG5, cell fractionation analyses of RFP-ATG4a and YFP-ATG18a showed that both proteins still associate with membranes in autophagy-inducing conditions when PI4K activity is inhibited by PAO (Fig. 5c–f). These results suggest that the absence of PI4P does not prevent the recruitment of early ATG proteins to the PAS/early phagophore where they likely accumulate as a result of a defect in AP formation.

To visualize where ATG5, ATG4, and ATG18 localize within the cell when PI4K activity is inhibited, we performed confocal microscopy analyses using the same conditions (lines, autophagy induction, and PAO treatment) than for cell fractionation analyses. As previously described, ATG5 and ATG18 were mostly found in the cytosol in rich conditions and formed puncta representing phagophores upon autophagy-inducing conditions[43,47] (Fig. 5g, h, −NC, DMSO). RFP-ATG4 presented a similar punctate signal in nutrient starvation conditions; likely representing autophagy-related structures as we found that RFP-ATG4a and RFP-ATG4b co-localizes with GFP-ATG8a in transgenic lines co-expressing the two constructs (Supplementary Fig. 6a–c). Upon PAO treatment, we observed a drastic decrease in the number of puncta for all proteins, i.e., ATG5-mCherry, YFP-ATG18, RFP-ATG4a, and RFP-ATG4b which, instead, showed a diffuse signal throughout the cytosol (Fig. 5g, h and Supplementary Fig. 6d). This result may seem contradictory with the membrane-associated accumulation of the proteins revealed biochemically (Fig. 5a–f); yet given the function of ATG8 in phagophore expansion[40] and the fact that PAO inhibits ATG8 lipidation, we propose that, in the absence of PI4P, phagophore expansion gets stalled very early causing the accumulation of premature membrane structures that are dense but below the diffraction limit hence the apparent cytosolic localization. Similar observations were made in the *atg13* mutant where, although ATG8 is lipidated and accumulates at the membrane, AP formation is blocked and the ATG8 signal looks diffuse in the cytosol[42]. Altogether our data suggest that the inhibition of PI4P synthesis does not prevent initial events in the nucleation of the PAS/early phagophore at which early ATG proteins accumulate, but is required for the proper assembly and expansion of the phagophore, notably by promoting ATG8 lipidation.

## APs form at PI4P enriched structures

PI4P is a major anionic phospholipid at the PM of plant cells where it acts as a docking site to recruit specific set of proteins via electrostatic interactions[18]. Our results show that PI4K activity is required for autophagy and proper phagophore expansion upon nutrient starvation in root cells. To test whether PI4P is directly present in the autophagic membranes we explored the localization of a set of PI4P-specific probes (mCitrine-1xPH[OSBP], mCitrine-2xPH[FAPP1] or mCitrine-1xPH;[FAPP1-E50A-H54A][18,20]) in regard to that of markers of autophagy

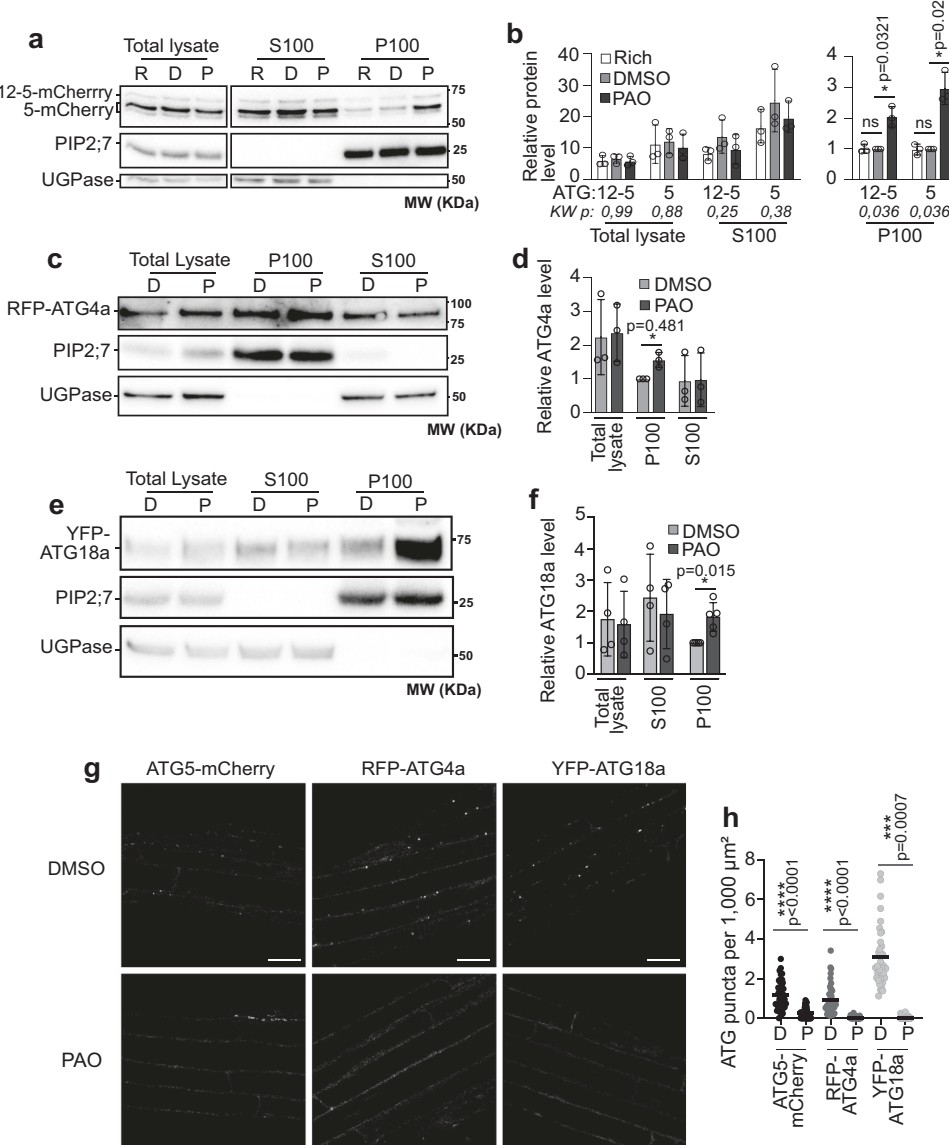

**Fig. 5 | PI4K inhibition prevents the formation of punctae-like structures of early ATG markers, but not their association to membranes.** ATG5-mCherry, ATG12-ATG5-mCherry (**a**), RFP-ATG4a (**c**), and YFP-ATG18a (**e**) accumulate in the membrane fraction in PI4P inhibition conditions. Immunoblot analyses using anti-mCherry antibody (**a**), anti-RFP antibody (**c**), and anti-YFP antibody (**e**). Uncropped blots in Source Data. 7-day-old seedlings were transferred from rich conditions (R) to nutrient deprived control conditions (D) or nutrient deprived medium supplemented with PAO 60 μM (P), for 30 min. Roots were dissected and proteins were analyzed by western blot after cell fractionation. PIP2;7 was used as loading control and normalization factor for the membrane fraction; UGPase was used as loading control and normalization factor for the soluble fraction. S100, soluble fraction; P100, pellet membrane containing fraction. Quantification of the levels of ATG5-mCherry and ATG12-ATG5-mCherry (**b**), RFP-ATG4a (**d**), and YFP-ATG18a (**f**) in conditions presented in **a**, **c**, **e** relative to P100 in DMSO conditions which was set to 1 in each independent experiment. Results present the average ± SD; number of independent experiments: $n = 3$ in (**b**) and in (**d**), in (**f**) $n = 5$ for P100 samples, $n = 4$ for other conditions. In (**b**) multiple comparison statistical analyses using the Kruskal–Wallis (KW) test were performed to assess differences among the 3 conditions (+N, Rich, PAO) in each group of samples (TL, S100, P100). Resulting exact $p$ values are indicated below the graph and only showed significant differences among the P100 samples. Follow-up statistical analyses comparing Rich or PAO conditions to DMSO were performed using two-tailed one sample t-test. In **d** and **f** statistical differences between DMSO and PAO conditions were performed using two-tailed one-sample t-test. ns, non-significant. **g** Confocal microscopy images of ATG5-mCherry, RFP-ATG4a, and YFP-ATG18a-expressing plants. 7-day-old seedlings were transferred to liquid nutrient deprived control conditions (DMSO) or nutrient-deprived supplemented with PAO 60 μM (PAO), for 30 min. Scale Bar: 20 μm. **h** Quantification of ATG puncta in conditions presented in (**g**). Results as presented as number of puncta per 1000 μm², bar is mean. For ATG5 puncta, three independent experiments were performed with $n = 41$ images for DMSO and $n = 68$ images for PAO. For ATG4 puncta, 4 independent experiments were performed with $n = 46$ images for DMSO and $n = 62$ images for PAO. For ATG18 puncta, three independent experiments were performed with $n = 38$ images for DMSO and $n = 40$ images for PAO. Statistical differences between conditions were assessed using two-tailed unpaired t-test, exact $p$ values are provided in the source data file.

structures in various conditions. We performed two types of experiments and compared the signal of PI4P probes to that of autophagic structures. First, we induced autophagy using nutrient starvation for 2 or 5 h in control conditions (−NC+DMSO). In addition, to increase the number of autophagic structures and address the potential dynamics

of PI4P throughout autophagy structures, plants placed in nutrient starvation were treated with 1 μM concanamycin A (Ca), an inhibitor of vacuolar hydrolases which results in the accumulation of autophagic bodies inside the vacuole[17] (Fig. 6a). In nutrient starvation conditions (−NC 2 h, −NC 5 h), signal from the PI4P probes 2xPH^FAPP1 and

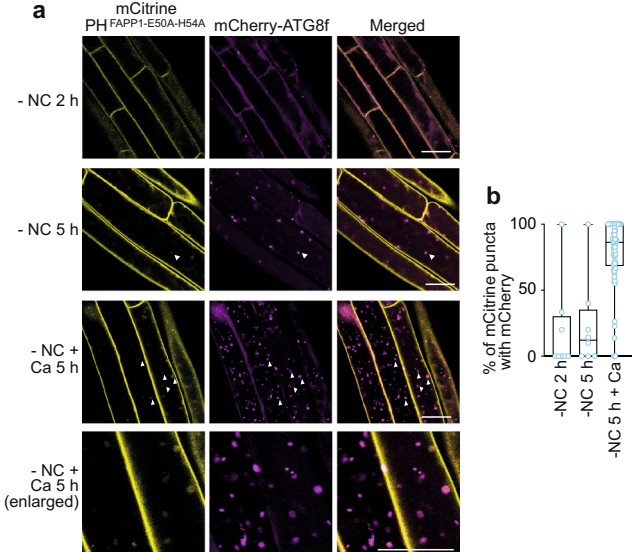

**Fig. 6 | PI4P accumulates on late autophagy structures. a** Confocal images of 7-day-old seedlings co-expressing mCherry-ATG8f and the PI4P-binding probe Citrine-1xPH$^{FAPP1\text{-}E50A\text{-}H54A}$. Plants were placed in liquid MS medium deprived of nutrients (−NC) for 2 or 5 h, in control conditions or supplemented with concanamycin A (Ca, 1 μM). Scale bar: 20 μm. **b** Quantification of the co-localization events in (**a**) as percentage of mCitrine puncta colocalizing with mCherry signal. Box plots indicate all individual values and median (middle line), 25th, 75th percentile (box), and 5th and 95th percentile (whiskers). For conditions −NC 2 h and −NC 5 h, $n = 8$ distinct replicates were examined over three independent experiments; for condition −NC 5 h + Ca, $n = 62$ distinct replicates were examined over five independent experiments).

1xPH$^{FAPP1\text{-}E50A\text{-}H54A}$ was mostly found at the PM with some endomembrane compartments corresponding to the TGN also observed (Fig. 6a, Supplementary Fig. 7), which is in accordance with previous publications[18,48]. In these conditions we observed very few punctate PI4P signal colocalizing with ATG8-labeled structures (Fig. 6a and Supplementary Fig. 7). The 1xPH$^{FAPP1\text{-}E50A\text{-}H54A}$ probe showed an overall stronger signal as well as more numerous colocalization events with mCherry-ATG8f (Fig. 6a, b). Nevertheless upon 5 h of autophagy induction, less than 10% of PI4P punctate structures displayed an ATG8f-mCherry signal (Fig. 6a, b, −NC). In contrast, when plants were treated with concanamycin A, we observed the accumulation of PI4P signal as dots inside the vacuole which strongly colocalized with ATG8 (80%) suggesting that PI4P labels autophagic bodies (Fig. 6, −NC+Ca). Together, these results suggest that there is little to no PI4P detectable on early autophagy structures but that PI4P accumulates on late autophagy structures likely through later fusion of APs with PI4P-enriched compartments such as endosomes as previously suggested[12].

While early ATG8 structures showing labeling for PI4P probes were rarely observed (i.e., in the cytosol), we noticed that a significant portion of ATG8 and ATG5 puncta were located on, or in close proximity to PI4P at the PM (Fig. 7a–d). Further, we observed a higher proportion of the phagophore marker ATG5 (compared to ATG8) adjacent to the PM concomitant with a reduced proportion of puncta in the cytosol suggesting that the early autophagy structures may preferentially be in contact with the PM. To gain quantitative and resolutive information into the cellular environment of AP formation we performed correlative light electron microscopy (CLEM) combined to Electron Tomography using plants co-expressing GFP-ATG8a and the PI4P probe mCherry-2xPH$^{FAPP1}$. Results presented in Fig. 7e, f and Supplementary Fig. 8 showed that upon autophagy induction, the vast majority of ATG8-labeled structures are found in the vicinity of the ER (<300 nm), and around 75% with less than 80 nm of distance suggesting direct contact with the ER membrane in roots of Arabidopsis

and supporting the idea that most, if not all, APs emerge from this compartment. In addition, we found a large proportion of ATG8 structures (around 40%) in the vicinity of the PM or the Golgi (<300 nm) and 20−30% in proximity to the mitochondria. We also observed plastids, vacuole or MVBs in the immediate environment or autophagy structures, albeit more rarely. To examine the relative distance of phagophores vs. APs to organelles, we counted open ATG8-labeled structures as phagophores and closed ATG8-labeled structures as autophagosomes although the latter group may contain a mix of both phagophores and APs given that a section of 150 μm-thickness cannot cover the whole phagophore/AP structure. In all cases, we found a larger number of phagophores vs. APs in close distance (<30 nm) to either the PM, the Golgi, or the mitochondria suggesting that, together with the contribution from the ER, these compartments, provide platforms for the phagophore assembly/elongation. Given the proximity of phagophores/AP with the PI4P-enriched PM, the accumulation of PI4P on late autophagy structures and the implication of PI4Kα1, the PM-localized PI4P producing enzyme, in autophagy (Fig. 2), we propose that the PM acts as a reservoir of PI4P for AP biogenesis.

## Discussion

AP formation is a highly dynamic process that requires fine tuning of several membrane remodeling steps. Lipids, aside from being structural membrane components are also known to modulate many membrane trafficking events in eukaryotic cells[16]. In that context, we speculated that lipids play critical roles in the formation of autophagic vesicles. Here, using both genetical and pharmacological inhibition of PI4P synthesis, we show that PI4K activity is important for AP formation and ATG8 lipidation suggesting that PI4P is required for the proper structuration and elongation of the phagophore. In contrast, our data suggest that PI4K is not required for the membrane association of earlier ATG proteins. Although we cannot exclude that these latter results are caused from unspecific effects of PAO, as these early ATG markers reside in the cytosol at steady state and are specifically recruited to the PAS upon autophagy induction, their recruitment to membranes in the absence of PI4P suggests that this lipid is not critical for the earliest steps in PAS formation and/or phagophore nucleation. Further, the absence of ATG-labeled puncta observed by fluorescent microscopy in these conditions (Fig. 5g, h) support the idea that PI4P is critical for the proper assembly and expansion of the phagophore. We further show that a significant number of phagophores are found in proximity with the PI4P-enriched PM (Fig. 7c–f; Supplementary Fig. 8). However, our data concerning the presence of PI4P on autophagic structures are equivocal, suggesting that PI4P is present on autophagic bodies (ie, the single-membrane autophagic structure delivered in the vacuole) but not, or at low levels, on the phagophore. These observations could be explained by different scenarios. First, PI4P might be mediating AP formation solely by contributing to PM-associated events; in that situation, the close proximity/contact of the phagophore with the PM may be sufficient to access the pool of PI4P required for AP formation. Second, if the concentration of PI4P on autophagic structures is low, PI4P-probes may not be recruited to these membranes, but rather titrated by highly-accumulating compartments such as the PM or the TGN[18]. Third, PI4P can be rapidly interconverted to other phosphoinositide species (PI, PI(4,5)P$_2$), and thus its presence in autophagic structures may only be transient, which could explain why we observed only a few colocalization events in the cytosol. Supporting the hypothesis of high dynamic turn-over of PI4P at autophagy structures, PI4P has been detected on autophagy structures in other organisms where it was proposed to play distinct roles at different steps of the process[49–52].

Accumulation of PI4P and its combination with other anionic lipids contribute to the identity and activity of membranes by mediating the selective recruitment of specific sets of proteins through

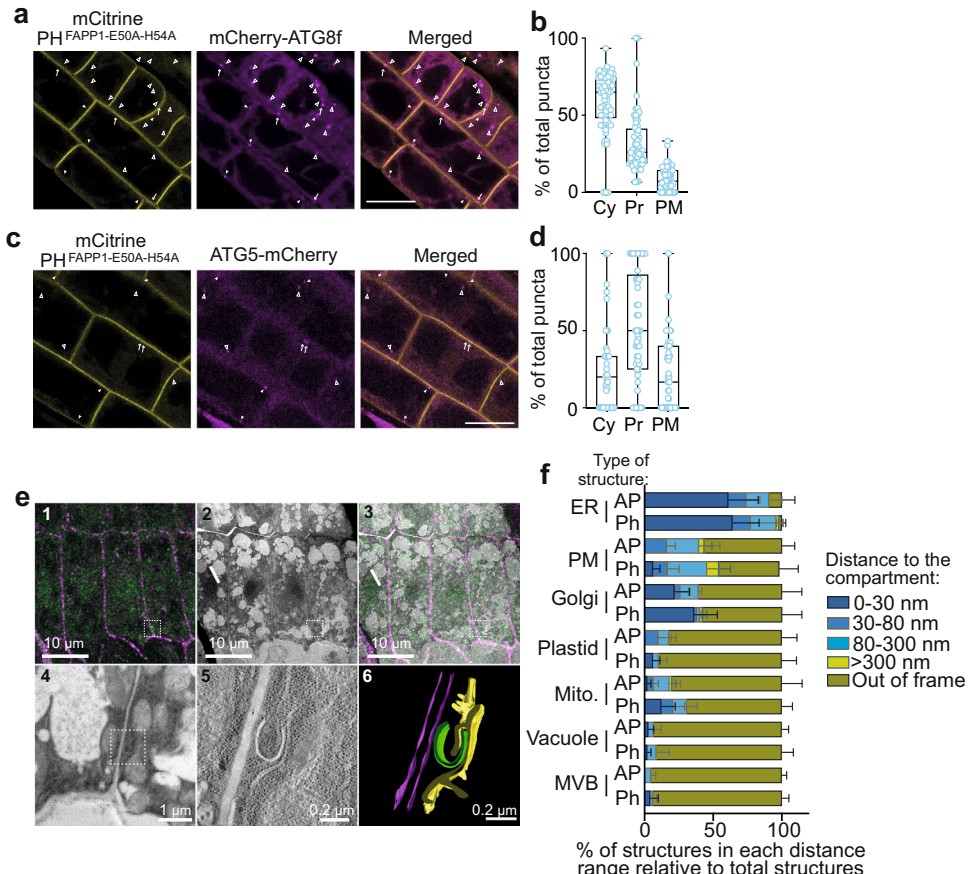

**Fig. 7 | Autophagy structures form close to PI4P-enriched compartments.** **a–d** Confocal images of 7-day-old seedlings co-expressing the PI4P-binding probe mCitrine-1xPH$^{FAPP1-E50A-H54A}$ and either mCherry-ATG8f (**a**) or ATG5-mCherry (**c**). Plants were placed in autophagy induction conditions (−NC) for 1.5 h. Empty arrowheads indicate puncta in the cytosol (Cy), full arrowheads indicate puncta at proximity from the plasma membrane (Pr) and arrows indicate puncta on the plasma membrane (PM). Scale Bar: 10 μm. Repartition of the three distinct types of ATG8 or ATG5 puncta populations are presented in **b** and **d**, respectively. For each image we counted the total number of puncta as well as the number of puncta in each category. We used these data to calculate the percentage of puncta in each category compared to total puncta. Box plots indicate all individual values and median (middle line), 25th, 75th percentile (box), and 5th and 95th percentile (whiskers). In **b**, n = 52 images were examined over four independent biological experiments; in **d**, n = 49 images were examined over 5 independent biological experiments. **e** (1) Confocal acquisition of the root tip of seedlings co-expressing GFP-ATG8a and 2xmCherry-2xPH$^{FAPP1}$ after 1.5 h induction of autophagy using 1 μM

AZD. (2) Transmission electron micrograph tiles of the regions observed in (1). (3) Correlated light electron microscopy view of the root tip which allows the precise identification of autophagic structures. (4) Magnification of a region of interest identified by CLEM allows to locate an autophagic structure and (5) to perform electron tomography to improve resolution and access to its 3D organization and cellular environment. (6) Segmentation of the autophagic structure. The plasma membrane is represented in magenta, the phagophore in green, the ER in yellow. **f** Quantification of (**e**). Proximity of ATG8-labeled structures (phagophores, Ph; autophagosomes, AP) with cell compartments in percentage of total phagophore or autophagosome analyzed. A total of 68 autophagy structures were analyzed over three independent biological samples. The distance of each structure to other cell compartments (ER, PM, Golgi, Plastid, Mitochondria (mito.), vacuole, or Multi Vesicular Body (MVB)) was measured, and structures were categorized in distinct range of distance compared to cell compartments. The results present the average ± SD of each category in % of the total structures analyzed in each biological sample (n = 3).

electrostatic interactions[16]. In that context, whether PI4P is directly present in autophagic membranes or at sites of the PM in contacts with the PAS from which the phagophores emerge, we propose that PI4P act as a docking site for the recruitment of a key protein or complex, working upstream of the lipidation of ATG8 to support the proper assembly and expansion of the phagophore. In both mammals and *Saccharomyces cerevisiae*, PI4P was shown to participate in autophagy, controlling ATG9 trafficking from the Golgi and/or TGN towards the PAS as well as regulating fusion of the APs with the vacuole/lysosome[49–52]. In addition, in mammals, PI4P is critical for the early steps of AP formation presumably by controlling the recruitment of ATG13, a member of the initiation complex[49]. Similarly, in Arabidopsis, our results show that a complete block in PI4P synthesis prevents early steps of AP formation, but also suggests fundamental differences concerning the role and/or effector(s) of PI4P in the pathway compared to other organisms. The Arabidopsis mutants *atg13* (knocked out for the initiation complex) or *atg9*, show no impairment in ATG8

lipidation[42,43] and the *atg9* mutant displays abnormally large and tubular ATG8-labeled structures[43]. However, our data show that blocking PI4P synthesis hampers ATG8 lipidation and the formation of ATG8-labeled structures (Figs. 3, 4 and Supplementary Fig. 5). These illustrate major differences in phenotypes caused by the absence of ATG13 or ATG9 compared to the lack of PI4P reported in the present work suggesting that, in Arabidopsis, PI4P does not function, or at least not only, by mediating the recruitment of ATG13 or trafficking of ATG9. Further, while ATG9 vesicles were shown to deliver the TGN-resident PI4KIIIβ to the PAS in mammals[49], here, we show that Arabidopsis mutants lacking the TGN-resident PI4Kβ are not affected in autophagy in our experimental conditions. Taken together, these suggest that, in Arabidopsis, PI4P functions prior to -and independent from- the implication of the ATG9 complex. Although we cannot exclude that PI4P also mediates ATG9-dependent mechanisms that were not uncovered in this study where the sharp block in PI4P synthesis blocked autophagy at a very early stage.

Previous work on plant autophagy showed that the ER is a critical platform for AP formation[43,47,53]. Our results support these previous conclusions as we found that the majority of autophagy structures are found in very close proximity to the ER (Fig. 7f). Additionally, we found that PI4P, a hallmark of the PM in Arabidopsis, is critically engaged in AP formation. In yeast and mammals, PI4P is found in two equivalent pools in the cell: at the Golgi and at the PM, and PI4 kinases from both the TGN and the PM participate in providing PI4P for autophagy[51,54]. In contrast, in Arabidopsis, PI4P mostly accumulates in the PM, with a secondary minor pool found at the TGN[18]. Our ET-CLEM analyses showed around 40% of autophagic structures in proximity with the Golgi (Fig. 7f) supporting the implication of the Golgi in the formation of, at least, a subpopulation of APs in nutrient starvation conditions as recently reported[55]. Nevertheless, our analyses of the TGN-residing PI4Kβ1 and PI4Kβ2 PI4P-producing enzymes show that these do not have a major effect on NBR1 degradation, plant survival to nutrient starvation (Fig. 2) or ATG8 lipidation and degradation (Supplementary Fig. 5). These data suggest that, in contrast to yeast and mammal models, these proteins are not involved in autophagy in plants (at least in our experimental conditions), or that their absence can be compensated by alternative proteins, including PI4Kα1. In contrast to PI4Kβ1 and PI4Kβ2, down-regulating *PI4Kα1*, which encodes for the PM-residing PI4K[28,30], reduces autophagy flux and largely affects plant resistance to nutrient stress (Fig. 2 and Supplementary Fig. 5). Further, our quantitative structural analyses of the cellular environment of AP formation showed that a substantial amount of autophagic structures are found close to the PM with a larger number of phagophores than AP in a short distance to the PM (Fig. 7e, f). Together, our results thus support that the PM provides a critical platform for AP formation and suggest that this compartment could functionally contribute to autophagy by providing the PI4P required for AP formation. A recent study suggested that ER-PM contact sites (EPCS) participate in autophagy: the relocalization of PM-localized proteins to EPCS is required to support AP biogenesis and mutants with altered EPCS show phenotypes reminiscent of autophagy deficiency[56]. Interestingly, PI4P has been shown to contribute to the formation of EPCS by recruiting tether proteins that allow the establishment of such sites[57]. Hence, a potential function for PI4P in AP formation might be to support the establishment of EPCS required for this process, potentially by mediating the recruitment of tether proteins at the site of AP formation. Additionally, or alternatively, PI4P could function directly in the phagophore membrane. It is to note that PI4Kα1 has been reported to localize to compartments other than the PM, including the nuclear membrane in protoplasts of Arabidopsis cells[58]. Thus, we cannot exclude a possible implication of PI4P from other pools than that at the PM, and further studies should explore the potential localization of PI4Kα1 directly at autophagy structures.

In sum, this study identified a critical component in autophagy in Arabidopsis: the lipid PI4P. Our results show that PI4K activity is critical for early stages of AP formation, that autophagy structures form closely to the PM and that the PM-associated PI4Kα1 is required for proper autophagy flux suggesting an important contribution of the PM to plant autophagy. We speculate that PI4P mediates the recruitment of a yet-unidentified PI4P-binding protein(s) that work(s) upstream of ATG8 lipidation to support the proper assembly and expansion of the phagophore, presumably at ER-PM contacts. Altogether, our study provides fundamental insights into the mechanism of plant autophagy and acknowledges the critical roles that lipids play in that process. Autophagy is a central program for plant acclimation to stresses; thus, by providing a better understanding of this mechanism this study contributes critical knowledge to help manage the effects of environmental changes on native and crop plants.

## Methods

### Arabidopsis lines

All experiments were performed in Arabidopsis of the ecotype Col-0, as wild-type, or derived of the ecotype Col-0. The following transgenic lines were used as previously described 35S::GFP-ATG8a[11], pUbi::GFP-ATG8e and pUbi::GFP-ATG8f[59], ATG5-mCherry[47], YFP-ATG18a[43], P7Y (1xPH^OSBP)[22], *pi4kβ1 pi4kβ2* (SALK_040479 SALK_098069[60]), *atg5-1* (SAIL_129_B07[11,61]).

To generate Arabidopsis lines stably co-expressing mCherry-ATG8f and PI4P-specific fluorescent probes, mCherry-ATG8f lines[53] were crossed with either P7Y (1xPH^OSBP), P21Y (2xPH^FAPP1), or with the mutated version of the FAPP1 probe (2xPH^FAPP1-E50A-H54A)[18]. For plants co-expressing ATG5-mCherry and PI4P probes; ATG5-mCherry[47] was crossed with the mutated version of the FAPP1 line (2xPH^FAPP1-E50A-H54A). To generate Arabidopsis lines stably co-expressing GFP-ATG8a and 2xmCherry-2xPH^FAPP1, the GFP-ATG8a line[11] was crossed with P21R (2xPH^FAPP1)[22]. For all aforementioned lines, the double homozygous F3 families were selected by Hygromycin B/Basta resistance.

TagRFP-ATG4a and TagRFP-ATG4b plants were generated as following. The ATG4a or ATG4b coding sequences were PCR-amplified using primers listed in Supplementary Table 1 and cloned into the pDONR™P2R-P3 plasmid by Gateway BP reaction. The gene sequences were verified by sequencing. The resulting plasmids were used for multisite gateway LR cloning together with the promoter UBQ10 cloned in the pDONRTMP4-P1R and tagRFP cloned in the pDONR221 resulting in the construct pUBQ10-tagRFP-ATG4a or pUBQ10-tagRFP-ATG4b in the expression plasmid PH7m34GW. The constructs were verified by sequencing and then transformed into *A. thaliana* Col-0 ecotype by floral dip; homozygous lines containing single construct insertions lines were selected based on Hygromycin B resistance.

Generation of *amiRNA:PI4Kα1* lines: two artificial microRNA sequences specific to the the class III *PI4Kα1* gene (*AT1G49340*) were designed using the software WMD3 Web MicroRNA Designer (ref. [62]; see Supplementary Table 1). The *amiRNA:pi4kIIIα1* precursor fragment sequences were PCR amplified from the pRS300 plasmid as previsouly described[62] using specific primers listed in Supplementary Table 1. PCR products were cloned into the pENTR/D-TOPO vector (pENTR Directional TOPO Cloning kit, Invitrogen). Clones were verified by sequencing and then transferred by gateway LR cloning into the destination vector PMDC7 which allows expression of the transgene under the control of β-estradiol[63]. The constructs were transformed into *A. thaliana* Col-0 ecotype by floral dip; homozygous lines containing single construct insertions lines were selected based on Hygromycin B resistance; 15 lines were obtained for each amiRNA. In order to test amiRNA efficiency, we compared the expression of *PI4Kα1* in each line compared to WT upon treatment with 10 μM β-estradiol in various conditions using RT-qPCR. From over 30 lines tested, only one line, corresponding to amiRNA sequence 1 (see Supplementary Table 1) showed a significant reduction in *PI4Kα1* mRNA and was therefore selected for further experiments.

### Growth conditions, autophagy induction, and chemical treatments

Seeds were vernalized in water and darkness, at 4 °C, for 24 to 48 h. Seeds were next surface sterilized in 10% bleach for 20–30 min, and sown on Murashige and Skoog (MS) agar medium plates (4,4 g.L⁻¹ MS powder including vitamins (Duchefa Biochemie M0222), 0.8% plant agar (Duchefa Biochemie, P1001), 1% sucrose (Merck Millipore, 84100) and 2.5 mM 2-(N-morpholino)-ethanesulphonic acid (MES, Euromedex EU0033), pH 5.7). Seeds were sown on a piece of mesh to facilitate root dissection from leaves. Seedlings were grown vertically for 7 days, at 21 °C, under long-day conditions (16 h-light/8 h-dark photoperiod, 300 μE m² s⁻¹). To induce autophagy, seedlings were starved for nutrients by transfer from MS plates into liquid MS −NC medium

(Murashige and Skoog liquid medium depleted for nitrogen and carbon: Murashige and Skoog micronutrient salts (Sigma, M0529), 3 mM CaCl$_2$ (Sigma, C5670), 1.5 mM MgSO$_4$ (Euromedex, P027), 5 mM KCl (Sigma, P9333), 1.25 mM KH$_2$PO$_4$ (Sigma P5655), 0.5% (w/v) D-mannitol (Sigma M9647), 3 mM MES, pH 5.7), and incubated in darkness (wrapped in aluminum foil) for different times, as indicated in the figures. For rich conditions, seedlings were sampled directly from the MS plates (in Figs. 3–5 and Supplementary Fig. 5) or incubated in rich liquid Murashige and Skoog (MS) medium (4,4 g.L$^{-1}$ MS powder including vitamins, 1% sucrose, and 2.5 mM MES, pH 5.7) under normal light conditions (in Figs. 1, 2b–d and Supplementary Fig. 4c, d). Chemical treatments were applied concomitantly with autophagy induction, as indicated in figures, as followed. To inhibit PI4kinase activity, seedlings were treated with PAO (Phenylarsine oxide, Sigma P3075) as previously described[18,64] with concentration ranging from 7.5 μM, 15 μM, 30 μM, or 60 μM and for different times as indicated in the figures. For PAO recovery assay (Supplementary Fig. 3), after DMSO or PAO treatment, seedlings were washed under agitation with 2 mL of liquid MS medium three times during five minutes, then three times during 30 min and a final long wash overnight. Five seedlings from both conditions were then transferred side by side to MS agar plates; 14 independent plates with a total of 70 seedlings were prepared over two independent biological experiments. Seedlings were grown vertically for seven days at 21 °C under long-day conditions. Pictures of the seedlings were taken on the day of transfer to MS agar plate (day 0) and seven days after transfer (day 7). Root length (day 0 and day 7) and rosette diameter (day 7) measurements (70 replicates over two independent repetitions) were determined using the Image J 1.8.0_172 software's (National Institutes of Health, USA, http://imagej.nih.gov/ij) straight or segmented plotting functionality. For TOR inhibition, AZD-8055 1 μM (MedChemExpress, HY-10422)[27] was used in liquid MS medium and for different times as indicated in figures. Concanamycin A 1 μM (Sigma-Aldrich, C9705) was used as previously described[17] as a vacuole acidification inhibitor to accumulate autophagic bodies in the vacuole. For all chemical treatments, DMSO (dimethyl sulfoxide, Sigma D8418) was used as vector control (untreated plants) in the same conditions (volume, time) than the treated plants. To induce the expression of the artificial micro-RNA against PI4Kα1 (AT1G49340), seedlings were incubated for 24 h with 10 μM β-estradiol (Sigma Aldrich, E2758). Plant survival to nutrient starvation was performed as described in Supplementary Fig. 2c. To assess pi4kβ1 pi4kβ2 survival compared to WT, seedlings were grown for 7 days on full MS plates with sucrose, seedlings were then transferred to a solid minimal medium lacking sucrose and nitrogen (medium was prepared as in ref. 65, without sucrose) for 7 days. Then, seedlings were transferred back to full MS medium with sucrose for an additional 10 days for recovery upon which survival rates were measured as the percentage of live seedlings (green) over total seedlings. 12 replicates were performed over three independent biological experiments with a total of 198 (WT) and 228 (pi4kβ1 pi4kβ2) seedlings tested. amiRNA:PI4Kα1 survival to nutrient starvation was compared to that of WT and atg5-1 mutants in conditions similar as the one stated above with the addition of 10 μM β-estradiol to induce amiRNA expression as described in Supplementary Fig. 2c. The experiment was repeated 20–22 times over four independent experiments with a total of 434 (WT), 480 (amiRNA:PI4Kα1) and 385 (atg5-1) seedlings tested.

## Microscopy analysis

All live confocal observation were performed on root epidermal cells of the elongation to early differentiation zone, employing 7-day-old seedlings mounted in culture medium with chemical treatments as indicated in the figures or in the following. Roots were kept under the microscope for no longer than 10 min in order to avoid secondary effects due to prolonged treatment times. Confocal images were acquired with a ZEISS LSM880 confocal system. Laser excitation lines for the different fluorophores were 488 nm for GFP or mCitrine and 561 nm for tagRFP, mCherry or propidium iodide. Fluorescence emissions were detected at 509–696 nm for mCitrine, 490–597 nm for GFP and 580–650 nm for tagRFP, and 639–709 nm for propidium iodide. In multi-labeling acquisitions, detection was in sequential line-scanning mode with a line average of 4. Confocal microscope images were processed using the Zen Black Software (Zeiss) for intensity optimization. High resolution Airyscan imaging of GFP and mCherry were performed using the ZEISS LSM880 confocal microscope equipped with a Plan-Apochromat 63×/1.4NA oil objective and a super-resolution AiryScan module with a band-pass filter 495–550 nm and a long pass filter 570 nm. Number of puncta, probe localization, cell number and surface areas were quantified using Zen Black Software (Zeiss) and Image J. Fluorescent puncta (ATG8, ATG18, ATG5, ATG4, and PI4P probes) were counted manually on a minimum of 12 images per condition and per independent experiment (representing over 80 cells total). At least three independent biological experiments were performed for each analysis. For propidium iodide assay, propidium iodide was added to the mounting medium at the working concentration of 10 μg/mL. For time lapses, time series were performed by acquiring 30 frames over 145 s.

For CLEM analyses, 5-day-old seedlings co-expressing 2xmCherry-2xPH$^{FAPP1}$ and 35S::GFP-ATG8a were incubated in liquid full strength MS supplemented with 1 μM AZD during 90 min to induce autophagy. High Pressure Freezing (HPF) and cryosubstitution, copper carriers (100 μm deep and 1.5 mm wide, Leica Microsystems, #16707898) were filled with −NC liquid medium supplemented with 20% bovine serum albumin (BSA) as a cryoprotectant. After induction of autophagy, root tips were rapidly cut and installed in the well of the carrier. Samples were frozen in the carriers using an EM-PACT high-pressure freezer (Leica) and then transferred at −90 °C into an AFS 2 freeze-substitution device (Leica). Samples were incubated in a cryosubstitution mix containing only uranyl acetate 0.1 % in pure acetone for 30 h (freeze substitution and embedding protocol adapted as in[66]). Afterwards temperature was raised progressively to −50 °C, at 3 °C h$^{-1}$. The cryosubstitution mix was removed and thoroughly washed 3 times with pure acetone and then 3 times in pure ethanol. Samples were then carefully removed from the carriers before progressive embedding in HM20 Lowicryl resin (Electron Microscopy Science): HM20 25%, 50% (2 h each), 75% overnight (diluted in pure ethanol), HM 20 100% (2 h) twice before a last 100% for 8 h. The polymerization was done under UV light for 24 h at −50 °C followed by 12 h at +20 °C. Samples in resin blocks were stored at −20 °C protected from light.

For microscope acquisitions and correlations, sections were made with an EM UC7 ultramicrotome (Leica). Sections of 150 nm were collected on parlodion coated 200-hexagonal mesh copper grids with thin bars (Electron Microscopy Science). Fluorescence Z-stack acquisitions were made on a Zeiss LSM 880 with excitation 488 nm for GFP and 561 nm for mCherry with a 63× apochromatic N.A 1.4 oil objective. The Z-stack were acquired to collect the maximum of photons emitted by the samples. Afterwards, maximal Z-projections were used to improve the visualization. Transmission electron microscopy observations were carried out on a FEI TECNAI Spirit 120 kV electron microscope equipped with an Eagle 4Kx4K CCD camera. Correlation was based on natural landmarks such as cell shapes. Correlated images were obtained using the plugin ec-Clem[67] of Icy software[68].

Tomogram acquisition and reconstruction were processed as in ref. 69. Image analysis was carried out using the ImageJ software (https://imagej.nih.gov/ij/download.html) with the Bio-Format plugin (https://www.openmicroscopy.org/bio-formats/) to open and process the Zeiss.czi files and to open the.mrc tilt series files and

tomograms. Distance between autophagic structures and other organelles was measure with ImageJ at the closest point between structures. Mosaic images of TEM were assembled using Photomerge function of Adobe Photoshop. No image analysis and measurements were carried out on mosaic and correlated images, they were only used as a visualization tools.

## GFP-ATG8 assay

For the GFP-ATG8 assay, total proteins were extracted from dissected roots exclusively. Roots were frozen in liquid nitrogen, disrupted using a TissueLyser (Qiagen), homogenized in 2X Laemmli buffer (5 µL/mg of fresh weight), and clarified by two consecutive centrifugations at 900 × $g$ for 1 min and 9600 × $g$ for 10 min, respectively. Proteins were denatured at 55 °C for 15 min. After migration on 12% SDS-PAGE (TGX Stain-Free FastCast Acrylamide kit, BioRad), equal protein quantity per lane was verified by stain-free activation of the loaded gel. After transfer to nitrocellulose (BioRad, #1704270), membranes were incubated with an anti-GFP mouse antibody (Roche, #11814460001) at a dilution of 1/5000. Peroxidase activity coupled to the Goat anti-Mouse antibody at a dilution of 1/5000 (Bio-Rad, #1706516) was revealed using Wester Lighting Plus-ECL (PerkinElmer) and detected with the Chemidoc MP Imaging system (Bio-Rad). Blots were imaged at different exposition times, long enough to observe clear distinct bands but prior to pixel saturation. Blot images were analyzed using the ImageJ software; band intensities were calculated using the plot surface areas functionality of the software. The intensities of the free-GFP band and the total GFP (GFP-ATG8 + free-GFP) were then used to calculate ratios representatives of the autophagic flux.

## Subcellular fractionation and protein analyses

For cell fractionation experiments, roots were dissected from seedlings after the treatments indicated in the figures. Roots were frozen in liquid nitrogen, disrupted and homogenized in the following buffer: HEPES 50 mM pH 7.5, Sucrose 0.225 M, MgCl$_2$ 2.5 mM, DTT 0.5 mM, PVP (Sigma P2307) 0.25% [w/v], PMSF 1 mM, antiprotease mix (cOmplete™ Protease Inhibitor Cocktail). Homogenates were filtered through a Miracloth layer and centrifuged twice at 1600 × $g$, for 15 min at 4 °C, for debris elimination. Clarified lysates (total lysate, TL) were centrifugated for 20 min at 137,000 × $g$ in the "Sorvall Discovery M150 SE Ultra-Centrifuge" (Hitachi), using the S100 AT6-0257 rotor to separate the membrane fraction (pellet, P100) containing membrane-anchored proteins from the soluble fraction (supernatant, S100). Pellets were resuspended in homogenization buffer (P100). To analyze samples from subcellular fractionation, the protein concentration of each sample (TL, P100, S100) was determined using Bio-Rad Protein Assay Dye Reagent Concentrate, (BioRad, ref #5000006) and measuring sample absorption at 595 nm. Equal amounts of proteins were prepared and denatured using Laemmli buffer, loaded onto 10–12% SDS-PAGE and analyzed by western blot as described above.

ATG5-mCherry was detected using a HRP anti-RFP antibody (Abcam, ab34767; 1/1000). YFP-ATG18 was detected using anti-GFP (Roche ;1/2000). Peroxidase activity coupled to the Goat anti-Rabbit antibody at a dilution of 1/5000 (Bio-Rad #1721019) was used for revelation of YFP-ATG18. tagRFP-ATG4 was detected using a mouse anti-RFP (Agrisera, AS153028, 1/1000). Peroxidase activity coupled to the Goat anti-mouse antibody at a dilution of 1/5000 (Bio-Rad #1706516) was used for revelation of tagRFP-ATG4. Anti-PIP2;7 (Agrisera, AS09 469, 1/5000) and anti-UGPase (Agrisera, AS05086, 1/10000) antibodies were used as loading controls and markers of the membrane fraction (P100) and soluble fraction (S100), respectively. Peroxidase activity coupled to the Goat anti-Rabbit antibody at a dilution of 1/5000 (Bio-Rad #1721019) was used for revelation of both PIP2;7 and UGPase.

To assess the level of ATG8 lipidation, total lysates (TL), soluble (S100) and membrane fractions (P100) were prepared as described

above. Equal amounts of proteins were loaded onto 13% SDS-PAGE gel containing urea (40% Acrylamide, 6 M Urea). ATG8 and ATG8-PE were detected using an anti-ATG8 antibody (Agrisera, AS142811, 1/1000). Extracts from WT plants were compared to that of the *atg5-1* mutant to clearly identify the ATG8-PE band, absent from the *atg5-1* mutant, unable to lipidate ATG8. ATG8 and ATG8-PE levels were quantified using PIP2;7 for the membrane fraction and UGP signals for the cytosolic fraction for normalization.

To assess NBR1 levels, 7-day-old seedlings were transferred from full MS plates into either rich MS liquid medium (+N), −NC liquid medium (−NC), or −NC liquid medium supplemented with 1 µM Concanamycin A for 16 h for *pi4kβ1 pi4kβ2* lines compared to WT. To test the impact of PI4Kα1 on NBR1 degradation, 6-day old seedlings of the *amiRNA:PI4Kα1* line, WT and *atg5-1* mutant, were first transferred to full MS plates supplemented with 10 µM β-estradiol for 24 h prior to being tranfered to liquid +N, −NC, or −NC+concA for 16 h, each supplemented with 10 µM β-estradiol (see Supplementary Fig. 2b). Upon treatments, proteins from total lysates were prepared as described above, loaded onto 10% SDS-PAGE and analyzed by immunoblot using an anti-NBR1 antibody (Agrisera, AS142811, 1/5000; upper part of the immunoblot) and an anti-H3 (Agrisera, AS10710, 1/5000e; bottom part of the immunoblot) and a peroxidase-coupled Goat anti-Rabbit secondary antibody as described above. NBR1, H3, and total protein signals were measured using ImageJ as described above. NBR1 levels were quantified using signals from total protein (stain-free) and H3 for normalization. For each sample, *NBR1* transcripts were analyzed in parallel by RT-qPCR as described in the next section.

## RT-qPCR analysis

Total RNA was extracted using the RNeasy mini kit (Qiagen). To eliminate genomic DNA contamination, an additional DNase treatment was performed according to the RNeasy kit instruction with the DNA removal kit (Invitrogen). One microgram of total RNA was reverse-transcribed into cDNA in a 20 µL reaction mixture using the Superscript II reverse transcriptase enzyme (Invitrogen). cDNA levels were then analyzed using the iQ™ Sybr Green supermix (BioRad) on the iQ iCycler thermocycler (BioRad) with the gene-specific primers listed in Table S1. The thermocycling program consisted of one hold at 95 °C for 10 min, followed by 40 cycles of 15 s at 95 °C and 1 min at 60 °C. After completion of these cycles, melting-curve data were then collected to verify PCR specificity, contamination, and the absence of primer dimers. The transcript abundance in samples was determined using a comparative threshold cycle method. The relative abundance of the reference mRNAs of *ACT2/8* and *AT4G33380* was determined in each sample and used for normalization according to the method described in ref. 70.

## Statistical analyses

Statistical analyses were performed using GraphPad Prism 9.3.1 (GraphPad Software, La Jolla, CA, USA) or using Microsoft Excel with tests indicated in the figure legends. $P$-values are as follows: $P$-value > 0.05 (non-significant, ns), $*P < 0.05$, $** P < 0.01$ and $*** P < 0.001$.

## Reporting summary

Further information on research design is available in the Nature Research Reporting Summary linked to this article.

## Data availability

Source data are provided with this paper.

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

## Acknowledgements

We thank Dr. Frédéric Domergue for critical revisions of the manuscript, Professor L. Jiang (The Chinese University of Hong Kong) for YFP-ATG18 seeds, Dr. Y. Dagdas (GMI, Vienna) for the pUbi::GFP-ATG8e and pUbi::GFP-ATG8f seeds and Dr. Satiat Jeunemaitre for the ATG5-mCherry seeds. We also thank the Bordeaux Imaging Center, part of the France BioImaging national infrastructure grant no. ANR-10-INBS-04. This project has received funding from the European Research Council (ERC) under the European Union's Horizon 2020 research and innovation program (grant agreement No 852136 to AB) and from Idex Bordeaux (AutoLip Emergence Program to AB).

## Author contributions

R.E.G., C.C., J.L., L.B., and J.J. designed and performed experiments and participated in the redaction of the manuscript. S.P. and J.C. performed experiments. L.N. and Y.J. provided critical material and reviewed the manuscript. AB designed and performed experiments, wrote and edited the manuscript.

## Competing interests

The authors declare no competing interests
