## [Peer Review File · Nature Communications]

Phosphatidylinositol-4-phosphate controls autophagosome formation in *Arabidopsis thaliana*REVIEWER COMMENTS

Reviewer #1 (Remarks to the Author):

Autophagy is a vital process for plant life, functioning in a variety of plant responses to stress and development. Although it is known that lipids play a major role in autophagosome formation and regulation, the actual lipid components of the autophagosome and their membrane origins are still relatively unknown. This is especially true in plant systems, for which autophagy research is still lagging. The current manuscript identifies PIP4 as a component of autophagosome formation in Arabidopsis plants and PI4K α as the enzyme participating in that function. Chemical and genetic perturbation of PI4P formation downregulate autophagosome formation, presumably during later stages of ATG8 lipidation rather than during autophagy initiation. This is very interesting, as PI4P participates in autophagy in mammalian and yeast systems, albeit at different stages of the process. The manuscript increases our knowledge regarding the lipid origins of autophagosomes, as well as the regulation of their formation. The experimentation is solid, and the text is well-written.

Major comments:

- The main issue that was not clear to me was the execution of nutrient starvation. It seems that in some cases, liquid medium was used and, in others, solid medium. Also, in figure 2A the phenotype presented is that of nutrient starvation and recovery, rather than starvation alone. This should be better clarified in the methods and the text. In addition, nutrient starvation was executed by transferring the plants to a poor medium for short time periods. The actual transfer of the plants between plates/ media can cause a response. It would therefore be useful to examine autophagy induction and inhibition by PAO when the plants are transferred to rich medium, to rule out the effect of plant transfer.
- Figure 2B – In the right panel, it seems that ConA treatment did not affect NBR1 accumulation, which is a bit strange.
- Figure 2C – I am not sure I understand the normalization here. Was each line normalized to its own +N treatment? This is a bit confusing as in atg5-1 lines, there is a massive accumulation of NBR1, and this is not reflected in the graph. The authors should consider clarifying that.

Minor comments:

- Lines 274-277: mCherry is spelled differently in several places.
- Figure 5G – the pictures here are not the best quality. The authors should consider revising it.

Reviewer #2 (Remarks to the Author):

A review of the manuscript titled "Phosphatidylinositol-4-phosphate controls autophagosome formation in Arabidopsis thaliana" by Gomez et al.

In this manuscript, the authors have reported the importance of the lipid PI4P in autophagy using pharmacological and genetical approaches. And they have shown that the plasma membrane-localized PI4K α 1 is involved in autophagy and that substantial portions of autophagy structures emerge from the PI4P-enriched plasma membrane. Finally, authors unravel critical insights into the molecular determinants of autophagy, proposing a model whereby the plasma membrane provides PI4P to support the proper assembly and expansion of the phagophore thus governing autophagosome formation in Arabidopsis. I think it is interesting as a first paper reporting the relationship between PI4P and PM on autophagy in plants. However, I am confused about the experiments regarding nutrient starvation and the use of inhibitors, and whether PI4P supplies from PM. I have three major points described below.

First, the authors found that PAO is a potent autophagy inhibitor and suggest that PI4P synthesis is critical for autophagy in root cells. The authors showed confocal images and a graph showing that APs are not induced in the presence of PAO (Fig 3). However, the cell damage or death is suspected by

PAO, and the reaction is stopped? I want to see some evidence that those cells are alive. I request to supply movies showing intracellular motions such as cytoplasmic streaming even in the presence of PAO. It would also be good to have data showing that individual plants do not die even after DMSO and PAO are washed out and transferred to a normal medium plate.

From figure 6, the authors described "these results suggest that there is little to no PI4P detectable on early autophagy structures but that PI4P accumulates on late autophagy structures likely through later fusion of autophagosomes with PI4P-enriched compartments such as endosomes as previously suggested". The authors induced autophagy under nitrogen and carbon starvation in the experiment shown in Fig. 6 and detected autophagosomes with concanamycin A. If a lot of APs are induced under nutritional conditions using AZD as shown in Figure 7e and Supplementary Figure 3, can you check whether mCherry-ATG8f and the PI4P-binding probe Citrine-1xPHFAPP1-E 50A-H54A exist together or not?

Reviewer #3 (Remarks to the Author):

In this manuscript, the authors investigated roles of PI4P in plant autophagy. They first showed that a PI4K inhibitor, PAO, blocked autophagy flux. Then, they revealed that the plasma membrane (PM)-localizing PI4K α 1 rather than PI4K β 1/2 that function in the TGN/endosomes is involved in the autophagy process. PAO inhibits the formations of ATG8-labeled autophagosomes in fluorescence microscopy and ATG8-PE in western blotting. However, PAO did not inhibit membrane recruitment of early ATG proteins, despite the lack of their punctate formation in microscopic analysis. PI4P probes showed that there was only little PI4P on early autophagy structures. Finally, CLEM analysis showed that phagophore/autophagosomes are in close proximity with the PI4P-enriched PM, suggesting that the PM act as a reservoir of PI4P for autophagosome biogenesis.

Major comments

This study nicely demonstrates an essential role of PI4K, especially that produced by the PM-localizing PI4K α 1, in plant autophagy. The CLEM analyses further strengthen the importance of the PI4K on the PM by demonstrating that phagophores are formed near the PM and ER, although the data can be still elaborated. As the detailed mechanism, they propose that PI4K functions at the nucleation step after the recruitment of early ATG proteins to the PM, which I think is unclear.

In the 3D-CLEM analysis (Figure 7), the extent of proximity should be defined by setting certain distances between organelles or only contact events should be counted. I suppose one section of 150 μ m-thickness cannot cover whole phagophore/autophagosome structure, but if the opening part of isolation membrane is found, like Figure 7e, it is definitely identified as a phagophore, thus looking at the membrane expanding process. So, counting only such phagophores may well support their conclusion. Also, some more examples of 2D images of CLEM should be presented for example as supplemented figures

As for the results that early ATG proteins are recruited to the membrane fraction in the biochemical analysis while no puncta formation was found in microscopy under PAO, it seems that some unspecific events that is not relevant to the PI4K function in autophagy may happen. Though the authors discuss some in the result part, it appears not so important to draw the main conclusions.

Knockdown experiment of PI4K α 1 could provide critical data to distinguish it from the function of PI4K β 1/2. Thus, this experiment should be applied to the ATG8 lipidation and puncta formation of ATG8 and earlier ATGs.

Minor comments

Figure 2: why does NBR1 mRNA increase after concanamycin A treatment?

Figure 4a: Why doesn't Atg8-PE in P100 increase with nutrient deprivation?

Figure 5:

- There is no information on the scale bars in the legend.
- "...per root area (in xxx?? nm²), ..."

Figure 6:

- Some arrowheads appear to indicate wrong positions.
- Including "+NC" and "+NC +Ca" might be preferable.

Figure 7:

- Arrows and arrowheads should be put in the left and middle columns. Don't "full arrowheads" indicate puncta "at proximity to PM"?
- Describe detailed quantification methods for Figure 7. What is "n" of "n=40-60"? What is the size of area analyzed? There is no "n" for Figure 7f.

Supplementary Figure 3: "...per root area (in xxx?? nm²), ..."

Supplementary Figure 4a: label "PIP₂7" ?

Supplementary Figure 4b: What are the labels, a, b, and c in the bottom of bars

Supplementary Figure 6: use same name of PI4P markers as described in the text and main Figure.

Line 38 and 112 "... that a substantial portions of autophagy structures emerge from the PI4P-enriched plasma membrane.": This study does not show the emerging process of autophagy structures from the PM.

Line 293: Is the sentence "..., (2) actually accumulate in the membrane fraction in these conditions" required?

Line 316: Is the "PI4P" actually "inhibition of PI4P synthesis"? This sentence should be revised.

Line 430 "Our ET-CLEM analyses showed around 30% of autophagic structures in proximity with the Golgi or Golgi derived vesicles and the ER (Fig. 7f) ...": I cannot find the Golgi-derived vesicles in the text.

REVIEWER COMMENTS

First, we would like to thank the reviewers for taking the time to assess our work. Following the reviewers' remarks and suggestions we now propose an updated version of our manuscript with changes highlighted in the text as well as additional experiments and results. Notably, we clarified the execution of nutrient starvation in all experiments and ruled out the sole effect of plant transfer in autophagy induction (Supplementary Fig. 4a,b; Supplementary Fig. 5a,b). We showed that plants resume root and rosette growth after recovery from PAO treatment indicating that PAO does not lead to cell death (Supplementary Fig.3). Finally, we provided additional 2D images of CLEM and set distances between organelles to finely quantify the proximity of phagophores and autophagosomes to other cell compartments (Supplementary Fig.8; Fig.7f).

We believe that our additional experiments and the changes provided to the manuscript answer to the reviewers' comments and significantly increase the quality of our study.

Second, before to the point-to-point responses to the reviewers below we would like to point to particular changes that we made in the figures:

- In the previous versions of Fig. 2c and Supplementary Fig. 5d, quantification was reported as levels of NBR1 or ATG8-PE relative to each genotype in +N condition which was set to 1. We decided to change that normalization and to present the results relative to WT in +N condition which was set to 1. We chose that normalization according to reviewer#1's comment on Fig.2b and because we thought that it gives a better representation of the phenotype of *ami:pi4kα1* compared to WT in all conditions. Note that this does not change the results or significance of the differences that we reported in the previous version of the paper.
- In Fig. 5d, the level of ATG8-PE was normalized by the use of a membrane protein, PIP2;7. While reviewing the data for this figure, we realized that in 2 experiments out of 8, the level of ATG8-PE had been normalized by the level of another membrane protein (data set 1) or by the average level of PIP2;7 and another membrane protein (data set 2). For consistency between experiments, we removed data set 1 and quantified the level of ATG8-PE solely by the level of PIP2;7 in data set 2. This caused very minor changes in the figure and did not affect the overall results, significance and conclusions of these experiments.

Reviewer #1 (Remarks to the Author):

Autophagy is a vital process for plant life, functioning in a variety of plant responses to stress and development. Although it is known that lipids play a major role in autophagosome formation and regulation, the actual lipid components of the autophagosome and their membrane origins are still relatively unknown. This is especially true in plant systems, for which autophagy research is still lagging. The current manuscript identifies PIP4 as a component of autophagosome formation in Arabidopsis plants and PI4K α as the enzyme participating in that function. Chemical and genetic perturbation of PI4P formation downregulate autophagosome formation, presumably during later stages of ATG8 lipidation rather than during autophagy initiation. This is very interesting, as PI4P participates in autophagy in mammalian and yeast systems, albeit at different stages of the process. The manuscript increases our knowledge regarding the lipid origins of autophagosomes, as well as the regulation of their formation. The experimentation is solid, and the text is well-written.

Major comments:

- The main issue that was not clear to me was the execution of nutrient starvation. It seems that in some cases, liquid medium was used and, in others, solid medium.

We apologize for the poor clarity of our explanation concerning the methods for nutrient starvation: the text, legend and material and method section has been revised to improve clarity (lines 173-177, 240-241, 254-257, 565-568, lines 592). In all experiments except Fig. 2a, nutrient starvation was performed by transferring seedlings in liquid medium lacking nitrogen and sucrose in the obscurity. For Fig. 2a, nutrient starvation was performed in solid medium using plates lacking nitrogen and sucrose and in the obscurity. We hope that our added explanations improve the understanding of our experimental conditions.

Also, in figure 2A the phenotype presented is that of nutrient starvation and recovery, rather than starvation alone. This should be better clarified in the methods and the text.

The text initially said, line 171: "To assess the function of *PI4K α 1* in plant physiology, we thus compared the recovery of the *amiRNA:PI4K α 1* and the *pi4k β 1 pi4k β 2* lines to that of WT plants as well as of the *atg5-1* mutant after 7 days of nutrient starvation (Supplementary Fig. 2c)". For better clarity we added additional explanation of the experimental procedure; see lines 173-177: "Briefly, 7-day-old seedlings were transferred from rich MS plates (+N) to MS plates lacking nitrogen and carbon (-NC) and placed in the obscurity for 7 days to induce autophagy. Plants were then transferred back to +N MS plates for 10 days to recover (see Supplementary Fig. 2c for additional explanation of the experimental procedure)".

In addition, nutrient starvation was executed by transferring the plants to a poor medium for short time periods. The actual transfer of the plants between plates/ media can cause a response. It would therefore be useful to examine autophagy induction and inhibition by PAO when the plants are transferred to rich medium, to rule out the effect of plant transfer.

As now mentioned in the material and method section as well as in Fig. 1, Fig. 2b-d and Supplementary Fig. 4c,d, rich conditions correspond to plants transferred from +N plates to +N liquid medium. In these conditions, the observed induction of autophagy and effect of PAO/ mutation is thus directly caused by nutrient starvation, ruling out the effect of plant transfer. In contrast, in Fig. 3, Fig. 4, Fig. 5 and Supplementary Fig. 5, rich conditions correspond to plants collected directly from +N MS plates and were compared to starved plants which were transferred to -NC liquid medium. We agree with the reviewer that transfer between plates and liquid media can cause a stress-mediated induction of autophagy in addition to the one caused by nutrient starvation *per se*. To test the impact of a transfer for 30 minutes from +N plates to +N liquid compared to -NC liquid, we measured the number of ATG8 puncta and the level of ATG8 lipidation. The number of GFP-ATG8a puncta was indeed significantly increased upon transfer from +N plates to +N liquid but only mildly compared to the large induction of ATG8 puncta observed in the -NC liquid conditions (Supplementary Fig. 4a,b). In +N liquid conditions ATG8 lipidation was not significantly affected compared to +N solid medium and -NC liquid (Supplementary Fig. 5a,b lipidation).

We now provide these additional data and explanation in the manuscript. Given these results, the sharp reduction of ATG8 puncta or ATG8 lipidation in PAO treatment (Fig. 3, Fig. 4) and the results from Fig. 1 and Supplementary Fig. 3c,d in which the effect of plant transfer was ruled out, we consider that the effects of plant transfer are negligible for the conclusions of our experiments.

- Figure 2B – In the right panel, it seems that ConA treatment did not affect NBR1 accumulation, which is a bit strange.

We agree with the reviewer that the accumulation of NBR1 was not very clear from that blot. We now included another image which is more in accordance with the average of our 4-7 independent biological experiments as showed in the quantification (Fig. 2c).

- Figure 2C – I am not sure I understand the normalization here. Was each line normalized to its own +N treatment? This is a bit confusing as in *atg5-1* lines, there is a massive accumulation of NBR1, and this is not reflected in the graph. The authors should consider clarifying that.

We apologize for the confusion in the normalization of these results, for better clarity we now propose a revised version of the figure with all data normalized to WT in +N conditions, the values for *atg5* are quite hard to quantify as the signal is highly saturated, which explain the large error bars in Fig. 2c, right panel.

Minor comments:

- Lines 274-277: mCherry is spelled differently in several places.

We thank the reviewer for pointing out these typos, we apologize and have corrected to text.

- Figure 5G – the pictures here are not the best quality. The authors should consider revising it.

We now provide images with improved resolution/quality.

Reviewer #2 (Remarks to the Author):

A review of the manuscript titled "Phosphatidylinositol-4-phosphate controls autophagosome formation in *Arabidopsis thaliana*" by Gomez et al.

In this manuscript, the authors have reported the importance of the lipid PI4P in autophagy using pharmacological and genetical approaches. And they have shown that the plasma membrane-localized PI4K α 1 is involved in autophagy and that substantial portions of autophagy structures emerge from the PI4P-enriched plasma membrane. Finally, authors unravel critical insights into the molecular determinants of autophagy, proposing a model whereby the plasma membrane provides PI4P to support the proper assembly and expansion of the phagophore thus governing autophagosome formation in *Arabidopsis*. I think it is interesting as a first paper reporting the relationship between PI4P and PM on autophagy in plants. However, I am confused about the experiments regarding nutrient starvation and the use of inhibitors, and whether PI4P supplies from PM. I have three major points described below.

First, the authors found that PAO is a potent autophagy inhibitor and suggest that PI4P synthesis is critical for autophagy in root cells. The authors showed confocal images and a graph showing that APs are not induced in the presence of PAO (Fig 3). However, the cell damage or death is suspected by PAO, and the reaction is stopped? I want to see some evidence that those cells are alive. I request to supply movies showing intracellular motions such as cytoplasmic streaming even in the presence of PAO. It would also be good to have data showing that individual plants do not die even after DMSO and PAO are washed out and transferred to a normal medium plate.

We agree with the reviewer that, although we used PAO as previously described in multiple publications, we had not tested the extent of its effect. According to the reviewer's suggestion we now provide evidence showing that plants show an altered development but do not die after PAO treatment. Supplementary Video 1 shows that cytoplasmic streaming is reduced but not completely abolished after 30 minutes of PAO treatment and we did not observe any sign of defects in cell integrity using propidium iodide in these conditions. In addition, Supplementary Fig. 3 shows that seedlings treated with PAO show no sign of death after 7 days of recovery post-treatment. Although we observed a reduction in root length and rosette

size of PAO-recovering plants after 7 days compared to control conditions, PAO-treated seedlings recovered and resumed root and rosette growth compared to the first day post-treatment showing that PAO treatment did not compromise cell/plant viability.

From figure 6, the authors described "these results suggest that there is little to no PI4P detectable on early autophagy structures but that PI4P accumulates on late autophagy structures likely through later fusion of autophagosomes with PI4P-enriched compartments such as endosomes as previously suggested". The authors induced autophagy under nitrogen and carbon starvation in the experiment shown in Fig. 6 and detected autophagosomes with concanamycin A. If a lot of APs are induced under nutritional conditions using AZD as shown in Figure 7e and Supplementary Figure 3, can you check whether mCherry-ATG8f and the PI4P-binding probe Citrine-1xPHFAPP1-E 50A-H54A exist together or not?

We performed these additional experiments (Supplementary Fig. 7c) but again did not see any clear co-localization between mCherry-ATG8f and the PI4P-binding probe Citrine-1xPH^{FAPP1-E 50A-H54A}.

Reviewer #3 (Remarks to the Author):

In this manuscript, the authors investigated roles of PI4P in plant autophagy. They first showed that a PI4K inhibitor, PAO, blocked autophagy flux. Then, they revealed that the plasma membrane (PM)-localizing PI4K α 1 rather than PI4K β 1/2 that function in the TGN/endosomes is involved in the autophagy process. PAO inhibits the formations of ATG8-labeled autophagosomes in fluorescence microscopy and ATG8-PE in western blotting. However, PAO did not inhibit membrane recruitment of early ATG proteins, despite the lack of their punctate formation in microscopic analysis. PI4P probes showed that there was only little PI4P on early autophagy structures. Finally, CLEM analysis showed that phagophore/autophagosomes are in close proximity with the PI4P-enriched PM, suggesting that the PM act as a reservoir of PI4P for autophagosome biogenesis.

Major comments

This study nicely demonstrates an essential role of PI4K, especially that produced by the PM-localizing PI4K α 1, in plant autophagy. The CLEM analyses further strengthen the importance of the PI4K on the PM by demonstrating that phagophores are formed near the PM and ER, although the data can be still elaborated. As the detailed mechanism, they propose that PI4K functions at the nucleation step after the recruitment of early ATG proteins to the PM, which I think is unclear.

We apologize for the confusion as it is not our conclusion that early ATG proteins are recruited to the PM. Instead, we propose that in the absence of PI4P, early ATG proteins still associate with the pre-autophagosomal structure (PAS) (see lines 312-314: 'These results suggest that the absence of PI4P does not prevent the recruitment of early ATG proteins to the PAS/early phagophore where they likely accumulate as a result of a defect in AP formation'). Our results show that when PI4P is absent, ATG proteins are still recruited to membranes. We presume that these membranes correspond to that of the PAS/early phagophore because that is where ATG proteins normally accumulates in response to autophagy induction and because confocal analyses do not show a mislocalization to other cell compartments in PI4P-depleted conditions (Fig. 5). Our biochemical analyses suggest that ATG proteins actually get stuck at this very initial compartment in the absence of PI4P, presumably because of its defects in organization which disrupts phagophore assembly/elongation as shown by both pharmacological and genetical analyses (Fig. 4, Supplementary Fig. 5). We therefore proposed that PI4P contributes to the proper structuration/elongation of the phagophore rather than PAS

assembly or phagophore nucleation *per se*. We clarified this point in the text lines 335 and 398-409.

Further, our data indicate that most phagophores are in contact with the ER and that a substantial proportion is found in close proximity to the PM (Fig. 7). This is in accordance with previous observations from Wang et al. (ref. 56), which suggested a contribution of ER/PM contact sites for AP formation. Therefore, we propose that, at least a portion of, the PAS forms in close relation to the ER and the PM.

In the 3D-CLEM analysis (Figure 7), the extent of proximity should be defined by setting certain distances between organelles or only contact events should be counted. I suppose one section of 150µm-thickness cannot cover whole phagophore/autophagosome structure, but if the opening part of isolation membrane is found, like Figure 7e, it is definitely identified as a phagophore, thus looking at the membrane expanding process. So, counting only such phagophores may well support their conclusion.

According to the reviewer's suggestion we measured the distance between 68 autophagy structures (phagophores vs. 'autophagosomes') analyzed by 2D CLEM and other cell compartments which are now presented in a revised version of Fig.7 and described in the text at lines 374-388. We thank the reviewer for this suggestion as it helped structuring our data and raised important points. First, it showed that most if not all autophagy structures, including both phagophores and APs are in proximity to the ER. Second, it showed the proximity of autophagy structures to the Golgi and the PM as well as, to a lesser extent, with the mitochondria. We also found contacts or proximity with other cell compartments but more rarely. Interestingly, a larger proportion of phagophores were found in closer distance (<30 nm) to the PM, the Golgi or the mitochondria compared to autophagosomes, which supports the idea that these compartments, with the additional major contribution from the ER, could provide platforms for the assembly or expansion of the phagophore.

Also, some more examples of 2D images of CLEM should be presented for example as supplemented figures

Following the reviewer's suggestion, we added additional 2D images of CLEM in Supplementary Fig. 8.

As for the results that early ATG proteins are recruited to the membrane fraction in the biochemical analysis while no puncta formation was found in microscopy under PAO, it seems that some unspecific events that is not relevant to the PI4K function in autophagy may happen. Though the authors discuss some in the result part, it appears not so important to draw the main conclusions.

We partially agree with the reviewer concerning that point. On the one hand, we agree that we cannot exclude that unspecific events related to the use of PAO causes the accumulation of earlier ATGs to membranes or the absence of puncta formation and we now mention this in our discussion/conclusions (lines 398-409). On the other hand, these results are not surprising and actually provide additional evidence of the absence of phagophore/AP that was observed using the GFP-ATG8 lines. Further, our results show that, similar to what was observed using PAO, genetically down-regulating PI4 kinase activity with the *ami:pi4ka1* line, also causes defects in ATG8 lipidation supporting that PI4P acts in phagophore expansion and AP degradation (Supplementary Fig. 5). As mentioned in the manuscript, this result is similar to what is found in the *atg13* mutant (ref. 42) where ATG8 lipidation is occurring and yet ATG8 does not form puncta in fluorescent microscopy. This shows that, when the formation of autophagy structures is stalled early on, they are not resolvable using confocal microscopy likely due to their small sizes and the weak ratio of the signal linked to the early structures versus the signal in the cytosol.

Knockdown experiment of PI4K α 1 could provide critical data to distinguish it from the function of PI4K β 1/2. Thus, this experiment should be applied to the ATG8 lipidation and puncta formation of ATG8 and earlier ATGs.

We agree with the reviewer that additional experiments could provide data to distinguish the function of PI4K α 1 from the function of PI4K β 1/2 thus finely deciphering the contribution of each pool of PI4P in autophagy. Nevertheless, we also believe that these additional experiments, which we anticipate to take up to 1.5 years to perform (accounting for the time-consuming generation of all required transgenics), fall out of the scope of the present study which main message is the implication of PI4P in autophagy with a larger contribution of PI4K α 1 than PI4K β 1/2. We hope that the reviewer agrees that the large variety of experiments already presented in the paper support this conclusion. Notably, experiments presented in the initial version of the manuscript showed that the knockdown of PI4K α 1 caused alterations to ATG8 lipidation and degradation (Supplementary Fig. 5c,d). In addition, and to address the reviewer's comment, we also performed these experiments on the PI4K β 1/2 double mutant and observed no difference in terms of ATG8 lipidation or degradation compared to WT (Supplementary Fig. 5c,d). Again, these results support our conclusion that PI4P provided by PI4K α 1 plays a major role in autophagy compared to that of PI4K β 1/2.

Minor comments

Figure 2: why does NBR1 mRNA increase after concanamycin A treatment?

We believe that these may result from some type of retro-control imposed by the blocking of autophagy or alteration of the vacuolar physiology.

Figure 4a: Why doesn't Atg8-PE in P100 increase with nutrient deprivation?

There is actually an increase in ATG8-PE when the level of the ATG8-PE band is normalized with the loading control PIP2;7 (see levels ATG8:PIP2;7 indicated underneath the blot). Additional experiments show that there is an increase on average over 4 independent experiments. Paired t-test indicates significant difference.

Figure 5:

- There is no information on the scale bars in the legend.

We apologize for all the mistakes and typos corrected below and we thank the reviewer for the thorough review of our manuscript. Information on the scale bars has been added in the legend.

- "...per root area (in xxx?? nm²), ..."

Information has been added on the root area in the figure and in the legend.

Figure 6:

- Some arrowheads appear to indicate wrong positions.

We corrected the positions of the arrowheads.

- Including "+NC" and "+NC +Ca" might be preferable.

There are very little ATG8 puncta in these conditions so less chances to see co-localization which is why we chose to not include these experiments that would not add critical information for the paper.

Figure 7:

- Arrows and arrowheads should be put in the left and middle columns. Don't "full arrowheads" indicate puncta "at proximity to PM"?

Arrowheads have been added to the left and middle columns. The reviewer is correct, full arrowheads indicated puncta at proximity to the PM, we apologize for this mistake that we corrected in the legend.

- Describe detailed quantification methods for Figure 7. What is "n" of "n=40-60"? What is the size of area analyzed? There is no "n" for Figure 7f.

We now provide additional information in the legend to explain the quantification methods for Fig. 7: '40 to 60 independent replicates (images) were analyzed in 4 to 5 independent biological experiments. For each image we counted the total number of puncta as well as the number of puncta in each category. We used these data to calculate the percentage of puncta in each category compared to total puncta. Results are presented as box plots with individual values of each replicate (n=40-60), min and max, bar is median.'

Supplementary Figure 3: "...per root area (in xxx?? nm²), ..."

This was corrected in the legend.

Supplementary Figure 4a: label "PIP₂?" ?

This was corrected in the figure.

Supplementary Figure 4b: What are the labels, a, b, and c in the bottom of bars

The labels a-c indicate different statistical groups. We now added information in the legend for better clarity.

Supplementary Figure 6: use same name of PI4P markers as described in the text and main Figure.

We used the same nomenclature for PI4P probes than in the main text or in the main figure in what is now Supplementary Fig. 7.

Line 38 and 112 "... that a substantial portions of autophagy structures emerge from the PI4P-enriched plasma membrane.": This study does not show the emerging process of autophagy structures from the PM.

We agree with the reviewer that this was a poor choice of words, as it is not our conclusion, or model, that the phagophore emerges from the PM. We have reformulated these sentences now line 38 (a substantial portion of autophagy structures are found in proximity to the PI4P-enriched plasma membrane), line 113 (autophagy structures are found in proximity to PI4P-enriched compartments) and line 427-428 (at sites of the PM in contacts with the PAS from which the phagophores emerge).

Line 293: Is the sentence "... (2) actually accumulate in the membrane fraction in these conditions" required?

According to the reviewer's comment we deleted that sentence for better fluidity of the text.

Line 316: Is the "PI4P" actually "inhibition of PI4P synthesis"? This sentence should be revised.

We thank the reviewer for pointing to that mistake, we have revised the sentence accordingly.

Line 430 "Our ET-CLEM analyses showed around 30% of autophagic structures in proximity with the Golgi or Golgi derived vesicles and the ER (Fig. 7f) ...": I cannot find the Golgi-derived vesicles in the text.

We now deleted the mention of Golgi-derived vesicles in the text and simply replaced by 'Golgi' which refers to both Golgi stacks and Golgi vesicles (line 460).

REVIEWERS' COMMENTS

Reviewer #1 (Remarks to the Author):

I have re-read the manuscript and the comments made by the authors. the authors have addressed all my concerns in a satisfactory manner and all my questions have been clarified. my one, very minor suggestion, is to change the word "obscurity" in line 174 to "darkness" or "dark conditions". I think this would be easier for non-native English speakers.

Reviewer #2 (Remarks to the Author):

The authors have adequately addressed our major concerns.

Reviewer #3 (Remarks to the Author):

I think the revised manuscript fully addressed my concerns.

REVIEWERS' COMMENTS

We thank the reviewers for assessing our revised manuscript.

Reviewer #1 (Remarks to the Author):

I have re-read the manuscript and the comments made by the authors. the authors have addressed all my concerns in a satisfactory manner and all my questions have been clarified. my one, very minor suggestion, is to change the word "obscurity" in line 174 to "darkness" or "dark conditions". I think this would be easier for non-native English speakers. We changed "obscurity" for "darkness" lines 174, 213 and in the legend of figure 2 (now line 996).

Reviewer #2 (Remarks to the Author):

The authors have adequately addressed our major concerns.

Reviewer #3 (Remarks to the Author):

I think the revised manuscript fully addressed my concerns.